



# The impact of urban land-surface on extreme air pollution over central Europe

Peter Huszar[1], Jan Karlický[1,3], Jana Ďoubalová[1,2], Tereza Nováková[1], Kateřina Šindelářová[1], Filip Švábik[1], Michal Belda[1], Tomáš Halenka[1], and Michal Žák[1]

[1]Department of Atmospheric Physics, Faculty of Mathematics and Physics, Charles University, Prague, V Holešovičkách 2, 180 00 Prague 8, Czech Republic
[2]Czech Hydrometeorological Institute (CHMI), Na Šabatce 17, 14306, Prague 4, Czech Republic
[3]Institute of Meteorology and Climatology, Department of Water, Atmosphere and Environment, University of Natural Resources and Life Sciences, Vienna, Gregor-Mendel-Straße 33, 1180 Vienna, Austria

**Correspondence:** P. Huszar (huszarpet@gmail.com)

**Abstract.** This paper deals with the urban land-surface impact (i.e. the urban canopy meteorological forcing; UCMF) on extreme air pollution for selected central European cities for present day climate conditions (2015-2016) using three regional climate-chemistry models: the regional climate models RegCM and WRF-Chem (its meteorological part), the chemistry transport model CAMx coupled to either RegCM and WRF and the "chemical" component of WRF-Chem. Most of the studies focused on change of average conditions or only on a selected winter and summer air pollution episode. Here we extend these studies by focusing on long term extreme air pollution levels by looking at not only the change of average values but also their high (and low) percentile values and we combine the analysis with investigating selected high pollution episodes too. As extreme air pollution is often linked to extreme values of meteorological variables (e.g. low planetary boundary layer height, low winds, high temperatures), the extreme meteorological modifications will be analyzed too. The validation of model results show reasonable model performance for regional scale temperature and precipitation. Ozone is overestimated by about 10-20 $\mu gm^{-3}$, on the other hand, extreme summertime ozone values are underestimated by all models. Modeled nitrogen dioxide ($NO_2$) concentrations are well correlated with observations, but results are marked with a systematic underestimation up to 20 $\mu gm^{-3}$. PM2.5 (particles with diameter < 2.5 $\mu gm^{-3}$) are systematically underestimated in most of the models by around 5 $\mu gm^{-3}$.

Our results show that the impact on extreme values of meteorological variables can be substantially different from that of the impact on average ones: low (5% percentile) temperature in winter responds to UCMF much more than average values, while in summer, 95% percentiles increase more than averages. The impact on boundary layer height (PBLH), i.e. its increase is stronger for thicker PBLs and wind-speed is reduced much more for strong winds compared to average ones. The modeled changes of ozone ($O_3$), $NO_2$ and PM2.5 show the expected pattern, i.e. increase in average 8-hour $O_3$ up to 2-3 ppbv, decrease of daily average $NO_2$ by around 2-4 ppbv and decrease of daily average PM2.5 by around -2 $\mu gm^{-3}$. Regarding the impact on extreme (95% percentile) values of these pollutants, the impact on ozone at the high-end of the distribution is rather similar to the impact on average 8-hour values. A different picture is obtained however for extreme values of $NO_2$ and PM2.5. The impact on the 95% values is almost 2 times larger than the impact on the daily averages for both pollutants. The simulated





impact on extreme values further well correspond to the UCMF impact simulated for the selected high pollution episodes.
Our results bring light to the principal question: whether extreme air quality is modified by urban landsurface with a different magnitude compared to the impact on average air pollution. We showed that this is indeed true for $NO_2$ and PM2.5 while in case of ozone, our results did not show substantial differences between the impact on mean and extreme values.

## 1  Introduction

More than 50% of the human population lives in cities and this number is expected to increase over 60% during the next 30
years (UN, 2018). Therefore, understating the impact of urbanization, i.e. the transition from rural to urban surfaces is crucial as there are evident consequences on the atmospheric environment affecting urban population (Folberth et al., 2015) which concerns both the climatic conditions (Chapman et al., 2017; Zhao et al., 2017), air pollution (Freney et al., 2014; Marlier et al., 2016; Im and Kanakidou, 2012) and possible interactions between them (e.g. Huszar et al., 2016b; Han et al., 2020).

It is now well understood that urban canopies, given their distinct geometric features covered with artificial materials (com-
pared to rural areas) influence meteorological conditions in a wide range of ways. Most importantly, the urban heat island (UHI) develops which means the accumulation of heat and its delayed release during night (Oke, 1982; Oke et al., 2017). Indeed, UHI is one of the most documented weather feature associated with urbanization affecting the temperatures of not only cities themselves but of entire surrounding regions (Huszar et al., 2014; Halenka et al., 2019) in dependence on the synoptic conditions (Žák et al., 2019). However, it is now well know that other meteorological parameters are perturbed too. Urban
land-surface is associated with decreased humidity as demonstrated recently by Marke et al. (2020) and in cities often the so called urban dry island (UDI) develops (Hao et al., 2018; Huszar et al., 2018a) with e.g. possible reducing consequences on fog formation (Yan et al., 2020). Another very important forcing that the urban canopy acts on the air in and above cities is caused by increased drag (Jacobson et al., 2015) and UHI induced lapse rate enhancement over cities (Karlický et al., 2018). The first influence manifests itself in a clear city-scale reduction of wind speeds (Zha et al., 2019). This drag further triggers mechanical
turbulence enhancing vertical mixing of scalars (Barnes et al., 2014; Ren et al., 2019; Li et al., 2019b) and consequently leads to elevated boundary layer height (PBLH; Flagg and Taylor, 2011). The second influence can on the other hand lead to urban breeze-like circulation (Ryu et al., 2013a; Ryu et al., 2013b; Zhong et al., 2017). In summary, the urban canopy layer forces the air within and above the canopy layer towards modified physical properties (temperature, humidity, windspeed etc.) and therefor we adopt here the term "urban canopy meteorological forcing" (UCMF) introduced by Huszar et al. (2020).
Not surprisingly, due to the UCMF the above listed changes in meteorological conditions have to lead to modifications in pollutant concentrations via modifying reaction rates, transport and deposition. Indeed, the presence of cities lead to perturbed air pollution not only due to the fact that they are responsible for release of large amount of gaseous and particulate pollutants (Seinfeld, 1989; Lawrence et al., 2007; Stock et al., 2013; Huszar et al., 2016a), but also due to UCMF. Many studies, both model- and observation based showed that UCMF causes decrease of average concentrations of primary pollutants like nitrogen
dioxide ($NO_2$), carbon monoxide (CO), sulfur dioxide ($SO_2$) and primary particulate matter (PM) (Struzewska and Kaminski, 2012; Chen et al., 2014; Kim et al., 2015; Huszar et al., 2018a, b; Ďoubalová et al., 2020). At the same time, UCMF can lead



to decrease in secondary pollutants like ozone $O_3$ due to removal of substances responsible for its destruction (Huszar et al., 2018a; Xie et al., 2016a, b).

While it has been clear that a very strong link must exist between air pollution, vertical eddy diffusion and, in general, the structure of the urban PBL (Masson et al., 2008), it is now shown too by many authors that the component that explains much of the UCMF induced concentration changes is the vertical eddy transport and its urban induced modifications (e.g. Wang et al., 2007, 2009; Zhu et al., 2015; Huszar et al., 2020). Using regional scale modeling techniques, Martilli et al. (2003); Sarrat et al. (2006); Struzewska and Kaminski (2012); Wang et al. (2007, 2009) showed that enhanced vertical eddy transport over cities results in decrease of primary gaseous pollutants ($NO_x$, CO) but leads to increase of ozone due to reduced titration. From the opposite direction, Fallmann et al. (2016); Han et al. (2020a) argue too that if mitigations in the form of roof greening or cool roofs were adopted, UHI would decrease along with decreased vertical turbulence which would turn into higher concentrations of primary pollutants and lower ozone. The dominant role of turbulence in increasing $O_3$ due to urbanization is stressed by Xie et al. (2016a, b) too. For particulate matter, the conclusions are similar to gaseous ones: enhanced vertical eddy transport lead to near surface reduction of both PM2.5 and PM10 (Zhu et al., 2017; Liao et al., 2014; Kim et al., 2015; Zhong et al., 2018). Li et al. (2019b) found that this acts mainly via the enhanced ventilation were the urbanization induced changes on wind play role too. Large-eddy-simulation (LES) approach was adopted by Li et al. (2019a) who concluded that vertical turbulence is a dominant process that determines the pollutant's removal from urban areas. A somewhat different behavior was encountered for primary- and secondary organic aerosol (POA/SOA) by Janssen et al. (2017): while POA responds to elevated turbulence by decrease, SOA will increase apparently due to enhanced downward transport from higher levels. Intermittent turbulence can play its role too in cities and can lead to rapid reduction of near surface particulate matter (Wei et al., 2018).

While the changes in concentrations due to UCMF presented by the listed studies are significant, they either looked at changes of averages values or changes during select short episodes (few days up to 1-2 months). From an air-quality perspective, much higher importance is attributed to changes in the high-end of the probability distribution of the modeled values, because extreme concentrations are more relevant regarding the health impact of air pollution in cities. This however, also requires to perform the analysis for a longer period than a few days or a selected month. In our previous studies that looked at the UCMF on air-quality (Huszar et al., 2018a, b, 2020), we were concerned on changes in values averaged over a 5 yr period, but the changes of extreme values remained unknown. Many of the studies listed (e.g. Struzewska and Kaminski, 2012; Ryu et al., 2013a; Ryu et al., 2013b; Zhu et al., 2015; Li et al., 2019b) analyzed episodes of high air pollution and often gave higher changes than our long term average values and this indicates that the high-end values of the distribution of modeled values is affected quantitatively in a different way. At the same time however, it was not clearly justified in these studies that the results are sufficiently robust and would hold for other episodes. Here we try to fill this gap and present a study that will look into the UCMF impact on ozone, NOx and PM2.5 near surface concentrations for a longer, 2yr period and instead of average values only, it will analyze the response of extreme values too which have a much higher policy relevance and may respond differently to the UCMF. Moreover, in line with many of the presented studies, it will pick selected high air pollution events too in order to demonstrate the UCMF impact in detail during these events. Finally, the study presented here adopts a multimodel approach in contrast with most of the studies listed. This is hoped to increase the robustness of the conclusions.





The paper consists of four main parts: after the Introduction, the models and their configuration, the experiments and the data used are described in the Methodology. In the Results section, simulations are first validated with respect to available meteorological and air quality measurements and then the changes in meteorological conditions and their subsequent impact on NOx, $O_3$ and PM2.5 average and extreme concentrations are presented. Finally, the results are discussed and conclusion are drawn.

## 2 Methodology

### 2.1 Models used

To describe the regional climate, the Regional Climate Model version 4.7 (RegCM4.7) and the Weather Research and Forecast with online chemistry version 4.0.3 (WRF-Chem) have been adopted. For regional air-quality (apart from the chemical model component of WRF-Chem), the Comprehensive Air-quality Model with Extensions version 6.5 (CAMx6.5) was used

RegCM4.7 is a non-hydrostatic mesoscale climate model being developed in the International Centre for Theoretical Physics (ICTP) (Giorgi et al., 2012). In our setup, the non-hydrostatic dynamic core wsa invoked . For convection, the Tiedtke scheme was chosen (Tiedtke et al., 1989). The cloud and rain microphysics is calculated with the explicit WSM5 5-class moisture scheme (Hong et al., 2004) while for radiative transfer, the Community Climate Model Version 3 (CCM3; Kiehl et al., 1996) approach was used. The turbulent transport of heat, momentum and moisture in the planetary boundary layer was parameterized using the non-local diagnostic Holtslag PBL scheme (HOL; Holtslag et al., 1990). Heat, radiation, momentum and moisture fluxes between the land-surface and the atmosphere are calculated within the Community Land Model version 4.5 (CLM4.5; Lawrence et al., 2011; Oleson et al., 2013) implemented in RegCM4.7. To resolve the meteorological phenomena associated with urbanized surfaces, the CLMU urban canopy module is implemented inside CLM4.5 (Oleson et al., 2008, 2010) which considers the classical canyon representation of urban geometry. Within the urban canyon, the Monin-Obukhov similarity theory with roughness lengths and displacement heights typical for the canyon environment is applied to calculate the heat and momentum fluxes (Oleson et al., 2010). Anthropogenic heat flux from air conditioning and heating is computed from the heat conduction equation based on the temperature inside of the buildings. Waste heat from air heating/conditioning is further added to the heat flux (Oleson et al., 2008).

WRF-Chem is a regional weather and climate model described in Grell et al. (2005). In the meteorological part of the model, the Rapid Radiative Transfer Model for General Circulation Models (RRTMG; Iacono et al., 2008) was used to predict long- and short-wave radiation transfer. The Purdue Lin scheme (Chen and Sun, 2002, PLIN;) is used for microphysics. Surface layer processes are resolved as in Eta model (Janjic, 1994) and land-surface processes are treated with the Noah land-surface model (Chen and Dudhia, 2001). Further, BouLac PBL scheme (Bougeault and Lacarrère, 1989), the Grell 3D convection scheme (only for low resolution; Grell (1993)) and the Single-Layer Urban Canopy Model (SLUCM; (Kusaka et al., 2001)) to account for the urban canopy effects are used.

In the chemical module of WRF-Chem that is online coupled to the main meteorological part, gas-phase chemistry is parameterized with Regional Acid Deposition Model, v. 2 (RADM2; Stockwell et al. (1990)), photolysis is resolved by Madronich





scheme (TUV; Madronich (1987)), aerosols are resolved by Modal Aerosol Dynamics Model for Europe and Secondary Organic Aerosol Model module (MADE/SORGAM; Schell et al. (2001)) scheme, together with simple wet deposition treatment (coarse parent domain only). MEGAN scheme (Guenther et al., 2006) was used for biogenic emission calculation, lightning-generated nitrogen oxides production is based on Price and Rind (1992). Wild fire, sea-salt and dust emissions are not considered.

Apart from the chemical module of WRF-Chem, chemical simulations were performed also offline with the chemistry transport model (CTM) CAMx version 6.50 (ENVIRON, 2018). CAMx is an Eulerian photochemical CTM that implements multiple gas phase chemistry schemes (CB5, CB6, SAPRC07TC). In this study, the CB5 scheme (Yarwood et al., 2005) was used. Particle matter concentration is computed using a static two mode approach. Dry and wet deposition are solved with the Zhang et al. (2003) and Seinfeld and Pandis (1998) methods, respectively. The ISORROPIA thermodynamic equilibrium model

(Nenes and Pandis, 1998) is also activated in our set-up to calculate the chemical composition and phase (partition between gas phase and condensate) of the ammonia-sulfate-nitrate-chloride-sodium-water inorganic aerosol system in equilibrium with gas phase precursors. Secondary organic aerosol (SOA) concentrations are computed with the SOAP equilibrium scheme (Strader et al., 1999).

CAMx is driven either with the WRF-Chem (i.e. its atmospheric part) or the RegCM model. To translate the meteorological

conditions from the driving model output to CAMx input, a meteorological preprocessor is needed: for WRF data, the wrfcamx preprocessor was used that is supplied along with the CAMx code http://www.camx.com/download/support-software.aspx. For RegCM, the preprocessor RegCM2CAMx originally developed by Huszar et al. (2012) was used. In both wrfcamx and RegCM2CAMx, the vertical eddy diffusion coefficients ($K_v$) are diagnosed following the CMAQ scheme (Byun, 1999) that was added to RegCM2CAMx in Huszar et al. (2016a). It is clear that the derivation of $K_v$ values follows here a different

concept than the PBL scheme of the parent models, however Lee et al. (2011) showed that using "non-consistent" method in calculating $K_v$ for CTMs does not imply less accurate results than coupling the PBL parameters directly. Cloud/rain/snow water content is taken directly from the parent models as in both models, the corresponding microphysics schemes (Purdue Lin and WSM5) provide explicit distribution of these quantities so their diagnostic derivation is not needed (in contrary to Huszar et al. (2011, 2012)). Given, that the coupling here is offline, no feedbacks of the modeled species concentrations on

WRF/RegCM radiation/microphysical processes were considered. Huszar et al. (2016b), using a similar setup than the coarse model here showed that the chemical perturbations induced by urban emission have a very small radiative effect in long-term average.

## 2.2 Model setup, data and simulations

Model simulations were conducted over a cascading nested domain configuration with the following horizontal resolution (and

size – as gridboxes): 9 km (189 x 165), 3 km (164 x 146) and 1 km (104 x 104). According to Tie et al. (2010), the threshold for the ratio of city size to resolution should be 1:6, which means 5 km or higher spatial resolution should be used to assess the chemistry of the cities we will focus (typical cities in central Europe – e.g. Prague, Berlin). Each computational domain is centered over Prague, Czech republic (50.075° N, 14.44° E) and uses the same map projection (Lambert Conic Conformal).



In vertical, the model grids are made of 40 layers in both RegCM and WRF-chem. The thickness of the lowermost level is
about 30 m and the model top is at 50 hPa (around 20 km) for each domain. Experiments were conducted for the 2014 Dec –
2017 Jan period with the first month used as spin-up and, additionally, for two short periods corresponding to high air pollution
event for the area of Prague based on the monthly reports of the Czech Hydrometeorological Institute (www.chmi.cz). These
periods were Feb 10 to Feb 23, 2015 with elevated PM2.5 pollution and Aug 2 to Aug 15, 2015 with elevated $O_3$. The long
period served to evaluate the long term UCMF and its impact on chemistry while the short episodes serve to demonstrate the
magnitude of this impact in detail during extreme air pollution.

The summary of all regional climate model (RCM) and chemistry transport model (CTM) simulations is given by Tab.1.
First, we performed experiments to analyze the urban canopy meteorological forcing over the 2 year period. These include
RegCM experiments on all three resolutions with offline nesting, and a 9 km WRF-Chem experiment. Apart from the default
RCM runs where urban surfaces were taken into account and parameterized with the urban canopy models mentioned above,
we performed in parallel experiments where urban surfaces were disregarded and replaced by a landuse type typical for the
surroundings for the urban area (most of the time "crops"). Accordingly, the runs with "urban" surfaces considered are suffixed
with "9U" (or "3U" or "1U", for higher resolutions) and those not considering them ("nourban") "9NU" (or "3NU" or "1NU").

After the regional climate model runs, we performed the CTM runs using CAMx. For the WRF-Chem runs this means
of course no additional experiments given its online coupled nature. CAMx runs performed using the RegCM meteorology
that considers and parameterizes urban landsurface are denoted "RegCM/CAMx9U" (or 3U/1U for higher resolutions) while
"RegCM/CAMx9NU" (or 3NU/1NU) denote simulations driven by "nourban" meteorology. Additionally, the 9 km resolution
WRF-Chem experiments serve as another CTM and finally, CAMx was driven also by the corresponding WRF-Chem mete-
orology denoted "WRF/CAMx9U" (or 9NU for the "nourban" case). Further, short term climate-chemistry experiments were
performed to demonstrate the UCMF impact on chemistry during extreme air pollution events. For these, WRF-Chem was run
in a nested mode similar to RegCM and denoted as "WRFchem9U' (or 3U/1U, and for the "nourban" case: 9NU/3NU/1NU).
Finally, these WRF-Chem runs served as driver for CAMx to obtain a further set of short term simulations: "WRF/CAMx9U"
(or 3U/1U) and "WRF/CAMx9NU" (/3NU/1NU) for the "nourban" case. With this complex design of experiments, we could
simultaneously investigate the long term urban impact (according to Huszar et al. (2014), 2 year is sufficiently long period for
significant urban impacts in models) while give the possibility to demonstrate the behavior of urban chemistry during high air
pollution periods.

The outer 9 km domain simulations were forced by the ERA-interim reanalysis (Simmons et al., 2010). The nested 3 and
1 km domains are forced by the corresponding parent domain using one-way nesting. Chemical boundary conditions for the
outer domains were taken from the CAM-chem data (Lamarque et al., 2012). Landuse information adopted in model simula-
tions was derived from the high resolution (100 m) CORINE CLC 2012 landcover data () and the United States Geological
190    Survey (USGS) database where CORINE was not available.

For European scale emissions, the TNO MACC-III (an update of the previous version II; Kuenen et al., 2014) data were
used. For the area of Czech republic, a high resolution national Register of Emissions and Air Pollution Sources (REZZO)
dataset issued by the Czech Hydrometeorological Institute (www.chmi.cz) and the ATEM Traffic Emissions dataset provided



by ATEM (Studio of ecological models; www.atem.cz) was used. The listed emissions data provide annual emission totals of
the main pollutants, namely $NO_x$, volatile organic compounds VOC $SO_2$, CO, PM2.5 and PM10. MACC-III data are gridded
data, while the Czech REZZO and ATEM datasets are defined as area, point and line (for road transportation) shapefiles of
irregular shapes that correspond to counties.

The original emission data from the listed emissions sources is preprocessed using the Flexible Universal Processor for
Modeling Emissions (FUME) emission model (Benešová et al., 2018, ; http://fume-ep.org/). FUME is intended primarily
for the preparation of CTM ready emissions files. As such, FUME is responsible for preprocessing the raw input files and
the spatial distribution, chemical speciation, and time disaggregation of input emissions. Emissions used are provided in 11
categories (SNAP – Selected Nomenclature for sources of Air Pollution) and category specific time-dissaggregation (van
der Gon et al., 2011) and speciation factors (Passant, 2002) are applied to derive hourly speciated emissions for CAMx and
WRF-Chem. Biogenic emissions of hydrocarbons (BVOC) for CAMx are calculated using the MEGANv2.1 emissions model
(Guenther et al., 2012).

## 3   Results

### 3.1   Model validation

Here we provide a basic comparison of the most important modeled quantities to measured data (for both the meteorology and
air-quality).

### 3.1.1   Meteorology

Fig. 1 presents the regional scale domain wide comparison of modeled and observed near surface temperature and precipitation
based on the E-OBS version 20.0e data (Cornes et al., 2018). RegCM shows overestimation of temperature, mostly during
winter months by up to 2-3 °C, especially over low lands. During JJA, the overestimation is slight higher, however, for the city
of interest, Prague, the model lies within +/- 0.5 °C. Over mountains, RegCM shows a systematic negative bias up to 3 °C (up
to 5°C for Alps). In case of WRF-Chem, temperature is rather underestimated by up to $1\,°C$ in both seasons and is in general
in better agreement with observation compared to RegCM. The overestimation of urban temperatures is caused due to fact that
E-OBS is interpolated and regridded from a relatively sparse network of stations unable to resolve local variation due to urban
effects (Kyselý and Plavcová, 2010).

Precipitation is slightly overestimated in RegCM, especially in winter by up to 1 $\mathrm{mmday}^{-1}$ in average. In JJA, the model-
observation agreement is better with biases within -1 to 1 $\mathrm{mmday}^{-1}$, larger positive negative bias occurs over western and
southeastern Europe. For WRF-Chem, winter is modeled with a fairly good agreement with biases within -0.5 to 0.5 $\mathrm{mmday}^{-1}$,
however model overestimates precipitation during JJA by up to 3 $\mathrm{mmday}^{-1}$.

Fig. 2 shows the model performance during the selected summer and winter periods for the most important meteorological
quantities controlling air chemistry: near surface temperature, 10-m wind speed and planetary boundary layer height. Regarding





temperature during the summer epizode, RegCM is able to capture daily maxima with a much higher success than WRF-Chem, in which case the model bias reaches -5°C. For daily nighttime minima however, RegCM shows large positive bias (1-2°C) while WRF-Chem experiments are more in line with observations. For the winter period, the model-observation agreement is highly dependent on the day. While during the first part of the episode, characterized by strong inversion and low daily maxima, models tend to overestimate the diurnal temperature range, while they agree better with measurements for day with

higher temperatures, when low level inversion clouds were dissipated. In general for winter episode, the agreement is better for WRF-chem experiments.

Wind is systematically overpredicted by both models in both episodes by about a factor of 2 for RegCM, while WRF produces, in general, even larger wind speeds. The correlation with observation is much lower than in case of temperature. One can also see, that higher observed wind speeds are captured with smaller bias.

Observation of the boundary layer height (PBL) here were deduced from ceilometer measurements, whose reliability depends on the meteorological conditions and have many shortcomings (Lee et al., 2019). Therefor instead of point-by-point comparison, we focus on the model biases in terms of maximum (minimum) daily boundary layer height. For the summer epizode, RegCM produces usually higher PBL heights (except a few days). The PBL in this model is set to reach a maximum possible value (about 3000 m), which is evidently reached during almost all analyzed summer days. The average maximum

PBL height in WRF-Chem experiments is around 2000 m, which means PBL height is slightly underestimated. During the winter episode characterized with low PBL height, its evolution is captured seemingly with a better accuracy for both models, while RegCM generates slightly larger PBL depths. A general behavior is that both models in both episodes tend to underestimate nighttime PBL heights connected to too stable stratification.

### 3.1.2  Air quality

Fig. 3 presents comparison of the modeled near surface concentrations to AirBase European Air Quality measurements (http://www.eea.europa.eu/data-and-maps/data/aqereporting-1) in terms of annual cycle of monthly means, diurnal cycle and histogram (probability density function; PDF) of daily means.

For ozone, there is a systematic overprediction of observed values for all models, while the RegCM driven CAMx simulations exhibit the largest bias up to 20-30 $\mu gm^{-3}$. Biases are smallest during summer months (almost zero for the WRF-Chem model)

while large overestimation occurs during the colder part of the year. According to the diurnal cycle, daily ozone maxima are reasonably captured with slight overestimation (underestimation) for RegCM (WRF) driven experiments and the timing of maximum ozone is somewhat shifted (by about 1 hour) in runs performed with CAMx. Large overestimation occurs during night, especially for RegCM driven CAMx runs (up to 40 $\mu gm^{-3}$) explaining the model bias during summer seen on the annual cycle. The histogram shows too that the distribution of modeled values has it maximum at larger values than the observed ones

(around 80-90 $\mu gm^{-3}$ compared to 60-70 $\mu gm^{-3}$ measured ). The low end of the measured distribution is poorly captured by all models.

$NO_2$ is underestimated by all models with a similar bias about 10-15 $\mu gm^{-3}$ during all seasons (with slightly lower bias during summer). The systematic underestimation holds, according to the diurnal cycles, even for each hours. However, the





model well correlates with measurements both in terms of the annual and diurnal cycles in both seasons. The underestimation

is well implied from the histograms too, with the most probable model values lying around 10 $\mu$gm$^{-3}$ while measured values has the most probable value about 30-40 $\mu$gm$^{-3}$.

Modeled PM2.5 concentrations are usually underestimated except the WRF driven CAMx runs (WRF/CAMxU9) in winter. All model setups well capture the annual cycle of PM2.5 with summer values underestimated by about 5-10 $\mu$gm$^{-3}$, especially in the WRF driven experiments. The diurnal cycle of PM2.5 in winter is characterized with two maxima resembling the

emissions, which is present in the modeled values too with a more pronounced amplitude. In general, the RegCM driven CAMx (RegCM/CAMx) experiments are closer to measurements than the WRF ones.

The model performance in terms of the daily maximum 8-hour ozone, NO$_2$ and PM2.5 near surface concentrations and the corresponding observations during the two selected extreme air pollution episodes is presented on Fig. 4. Models underestimate the high ozone concentrations with the best match for the RegCM driven CAMx experiment. Although RegCM/CAMx,

according to Fig. 3, slightly overestimates the daily maximum ozone in average, it is still unable to resolve extreme values as seen on the histogram. NO$_2$ is greatly underestimated during this episode, as expected from the general behavior seen on previous figure for all models. The models are poor in exhibiting this large negative systematic bias but also fail to capture the day-to-day variation. Exception is the 1 km WRF-Chem result, which could resolve the daily variation quite well. During the winter episode, PM2.5 is underestimated by WRF-Chem and RegCM driven CAMx runs, with the latter having smaller biases

around -10 $\mu$gm$^{-3}$. CAMx experiments driven by WRF meteorology tend to overestimate PM2.5 during this episode. NO$_2$ is, as expected from the pervious figure (and similar to summer), underestimated (mainly its peak during Feb 20-21) while high resolution experiments (RegCM/CAMx and WRFchem) have the tendency to simulate peak values during the early days of the episode. This can be connected to underestimation of the PBL height seen in Fig. 2.

## 3.2 Impact on meteorology

Before looking at how extreme air pollution events respond to the introduction of urban surfaces (i.e. to the UCMF), we present how the meteorological conditions driving these air pollution cases change due to urban landsurface. The most important three parameters will be analyzed that are a major part of the UCMF (Huszar et al., 2018a, b): the near surface temperature (tas), the height of the boundary layer (PBLH) and the 10-m wind speed (wind10m). Apart from the changes of the mean values, we will also look at the changes at the tails of the probability distribution function (PDF) of these meteorological quantities. Indeed,

extreme air pollution events are often related to high temperatures (high ozone episodes), low winds (stagnant conditions with limited dispersion from sources) and low boundary layer height (stable conditions with inversion layer(s) and very limited mixing).

### 3.2.1 Impact on average values

Fig. 5 shows the average JJA and DJF urbanization-induced-change of temperature, boundary layer height and 10-m wind

speed for both RegCM and WRF-Chem 2015-2015 experiemnts as the difference between the "urban" (U) and corresponding "nourban" (NU) experiments for the area of Prague (with indicated administrative boundaries). To ease the comparison of the





RegCM (performed in all three resolutions) performance with WRF-Chem (only 9 km), apart from the 1 km RegCM result, we also plot the result from the 9 km experiment.

Temperature is increased due to urban landsurface by more then 1 ° C in summer for the 9 km RegCM run while much

larger increase is resolved for the city center in the 1 km RegCM experiment (up to 3 ° C). WRF-Chem produces comparable increase around 2 ° C. For winter, the impact on temperature is weaker in RegCM compared to summer, except WRF-Chem, which produces again warming around 2 ° C. For the PBLH, the impact is larger during winter and most pronounced for the city center in the 1 km RegCM experiment (up to 300-400 m increase). In the 9 km experiments the increase is much smaller reaching 150 m and 300 m in summer and winter, respectively. Regarding the wind decrease, it is again most pronounced in the

1 km RegCM in the city center (around -1.5 $\mathrm{ms}^{-1}$ change in summer). The winter decrease is in general lower and the 9 km resolution runs produce lower wind decreases too (compared to the 1 km RegCM run), around -0.5 $\mathrm{ms}^{-1}$. The figure clearly demonstrates the importance of resolution with higher ones resolving the city center peaks of the impacts.

### 3.2.2 Impact on extreme values

Apart from the change of the average values, we are also interested in examining, how the above analyzed quantities change

in their tails of the PDF. Results are summarized in Tab. 2 for 4 cities, Prague, Berlin, Munich and Budapest and the 5% and 95% percentiles are analyzed (besides the mean values). The two numbers separated by "slash" mean result for the RegCM and WRF-Chem simulations. Results are from the 9 km experiments except for Prague, were for the RegCM experiment we took results from the 1 km domain.

During winter, the 5% percentiles exhibit a larger increase compared to the change of means, although this depends on the

model and city chosen. For Prague, the changes for the average values vs. 5% are 2.4 vs.5.0 ° C in the 1 km RegCM run, while the difference for other cities and models are lower (around 1.5 vs. 2.0 ° C). The change of the 95% percentile is lower than the mean change for RegCM experiments and, for Berlin and Prague, also for the WRF-Chem ones. A more consistent picture is achieved for the JJA changes. In all models and for all chosen cities, the change of the 5% value is lower than the change of the mean ones (0.5-2 vs. 1.2-2.4 ° C). On the other hand, the change of the 95% values is larger than the change of the mean

ones (1.5-3.0 ° C). In summary, in DJF low temperatures are increased stronger due to urban landsurface than the mean values while in JJA, high values increase even more while low values are modified less due to the introduction of urban landsurface. This means that during summer, the PDF for temperature is wider after the rural-to-urban transformation.

For the PBLH, in DJF the 5% percentile change (around 50-250 m increase) is lower in every city and model than the change of the mean values (roughly 150-350 m increase). High increases are characteristic for the the 95% percentile too (compared

to the mean values), around 150-450 m. The general behavior of the PBLH and its change due to urbanization is similar in summer. While the change of the 5% is somewhat smaller than the change of the mean values (120-250 m vs. 240-480 m), the high end of the PDF responds with slightly higher increases (250-490 m).

In case of windspeed decrease at 10-m in DJF, the low end of the PDF responds less than the mean values (around -0.3 $\mathrm{ms}^{-1}$ compared to -0.5 to -1 $\mathrm{ms}^{-1}$). On the other hand, the decrease at 95% is much larger, around (-0.7 to -2.5 $\mathrm{ms}^{-1}$). Qualitatively

a similar picture is seen for JJA although the urbanization induced changes are smaller. The change of the 5% value is again





around -0.3 ms$^{-1}$, i.e. less than the change of the mean ones (-0.3 to -0.5 ms$^{-1}$). On the other hand, the high end of the PDF corresponds to stronger decrease (-0.3 to -1.5 ms$^{-1}$). In summary, higher windspeeds are prone to larger decreases due to the drag induced by the urban landsurface.

### 3.3 Impact on the air-quality

The above presented meteorological changes (i.e. the UCMF) are expected to have implications in air pollutant concentrations and here, we will also focus on the change of extreme concentrations, i.e. we will be interested in the behaviour of the tails of the PDF. While from AQ perspective, the high end the PDF are of relevance, for completeness, we will also investigate the change of the low values. In particular, the 5% and the 95% percentiles will be analyzed along with the change of the mean values. Spatial figures show the change for Prague (with indicated administrative boundaries) and its surroundings. Result are

calculated as the difference between the corresponding "urban" (U) and "nourban" (NU) experiments.

#### 3.3.1 Impact on ozone

In Fig. 6 the UCMF impact on JJA average and 95% percentile daily maximum 8-hour $O_3$ (DMAX8HO3) is plotted for Prague. We are interested here, whether extreme values of DMAX8HO3 are impacted by the urban canopy meteorological forcing more than the mean values. As expected, the introduction of urban landsurface causes an increase of near surface

ozone concentrations. In the 1 km RegCM/CAMx experiment, the impact on mean is around 2-3 ppbv increase, while the 95% percentile is increased slightly more, by about 3-4 ppbv. The impact on mean values is similar for the 9 km RegCM/CAMx and WRF/CAMx experiments and for these model setups, the increase at the high end of the PDF is also around 3-4 ppbv. For the WRF-Chem experiment, again, extreme values of DMAX8HO3 increase more (around 4-6 ppbv) compared to the change of mean values (3-4 ppbv).

To extend our analysis to a larger number of samples for obtaining more robust results, we summarized the urban-canopy-induced absolute change of ozone in the centres of four cities: Prague, Berlin, Munich and Budapest and the results are presented in Tab. 3. Besides mean and 95% values, we included for completeness also the JJA change of low values (5% percentile). Regarding the change of the lower end of the PDF, the picture is not clear and both lower and higher changes with respect to the change of the mean values is encountered. For RegCM/CAMx, the change for 5% is usually lower, for

WRF/CAMx, it is clearly higher and for WRF-Chem, it is again rather lower than the corresponding change of the means. For the change of the high values (95%), for RegCM/CAMx and WRF-Chem, there is an indication that the high-end of the PDF responds to the UCMF with larger increase compared to the change of the mean values (2 to 4.5 ppbv vs. 2-3.5 ppbv.), however, in WRF/CAMx, the change 95% values tend to be rather lower. In relative numbers (Tab. 4), the increase of 5% values is clearly higher compared to the relative change of mean values and this holds consistently for each city and all model

simulations. On the other hand, the relative increase of the 95% values tends to be lower than the corresponding relative change of the mean values, and again, this holds for each city and every model.



### 3.3.2 Impact on $NO_2$

In Fig. 7 the UCMF impact on the DJF and JJA mean and 95% percentile daily mean $NO_2$ is plotted. The change of the average DJF concentrations is about -2 to -4 ppbv, being highest in the WRF-Chem experiment. Regarding the 95% percentile values, the change is evidently larger compared to the change of means. It is around -6 ppbv in both the 1 km RegCM/CAMx and WRF-Chem experiments, and somewhat lower, around -4 ppbv, in the rest of the simulations. In summary, results show an evident larger decreases of extreme $NO_2$ values compared to the decrease of average values.

We extend our analysis again for other cities and to the changes of the low-end of the PDFs too (see Tab. 5, upper part). Looking at winter 5% values, it is evident for each city and model that these are prone to smaller reduction due to urban effects compared to mean values (around -0.5 to -1 ppbv change vs. -1 to -7 ppbv, depending on the city/model). The change of 95% is on the other hand much larger (often 2 times) compared to the decrease of the mean values, in most of the cases ranging from -3 to -12 ppbv decrease (with the sole exception of Munich and the RegCM/CAMx run). The overall picture in JJA is qualitatively similar to the DJF case. While the decrease of mean values are between -1 to -7 ppbv, the decrease in case of the 5% percentile is about -0.5 to -3 ppbv, and, on the other hand, the decrease of the 95% values are much larger, lying between around -1.5 to -12 ppbv (largest in the WRF driven experiments, usually above -5 ppbv).

As the urban canopy meteorological forcing (UCMF) induced $NO_2$ changes are caused primary by vertical turbulence transport (Huszar et al., 2020), the amount of removed material (i.e. the $NO_2$) is expected to be proportional to the absolute amount of that material. This could explain the larger change for the high-end of the distribution. Whether this is true, or other non-linear feedbacks play role too, we also analyzed the relative change of the mean, 5% and 95% quantiles (as done for ozone too) and results are summarized in Tab. 6 (upper part). The relative change of the 95% values are shown only. For DJF, the relative change of the mean values is about -15 to -20%. For the 95% percentile change, the relative change is both larger and smaller depending on the city and model choice. Unlike in summer, the relative change of the 95% values is evidently higher than the relative mean change, especially in the WRF driven runs (WRF/CAMx and WRF-Chem) were it can exceed -50% change compared to the -30 to -40% change for the mean values.

### 3.3.3 Impact on PM2.5

In Fig. 8, similarly to $NO_2$, the UCMF impact on the DJF and JJA average and 95% percentile daily mean PM2.5 is plotted. In winter, the change of the mean values is around -2 to -4 $\mu gm^{-3}$ in the RegCM driven simulation up to -5 $\mu gm^{-3}$ for WRF driven ones for the center of Prague. The change of the 95% percentile is clearly larger: it reaches -6 $\mu gm^{-3}$ in every simulation except the 9 km RegCM/CAMx experiment. In JJA, the UCMF induced PM2.5 changes are smaller. The change of mean values is around -2.0 $\mu gm^{-3}$ (smaller only in the 9 km RegCM/CAMx experiment). Again, the change for the high-end of the distribution is much larger and evident in each model. For the high resolution RegCM/CAMx experiment, it reaches -4 $\mu gm^{-3}$ while around -2 to -3 $\mu gm^{-3}$ in other models. In summary, results show again evidently larger decreases of extreme PM2.5 values compared to the decrease of mean ones.




Extending our analysis again for other cities and also to the change of the low-end of the PDF (Tab. 5), wee can see in DJF,

that the 5% percentile changes are evidently lower than those of means, similar to $NO_2$. The 95% percentile values (decreases) are however much larger reaching -5 to -10 $\mu gm^{-3}$ compared to changes of means (reaching -3 to -5 $\mu gm^{-3}$). The JJA behavior is qualitatively similar to the DJF one: the changes of the 5% values are smaller in absolute sense compared to the change of means (up -0.1 to -1.5 vs -0.5 to -2.5 $\mu gm^{-3}$). Again, the 95% percentile changes are much larger compared to the change of means reaching -4 $\mu gm^{-3}$.

Looking at the relative changes in Tab. 6 (lower part) for DJF, there is an indication that the 95% percentile change is larger than the mean one in relative sense, although models are not unified and for some model experiments, the relative changes of 95% vs. mean values are rather similar. In JJA, the relative change of the 95% percentiles is however clearly much more higher than the corresponding change of the mean values, especially for the 1 km RegCM/CAMx experiment. In summary, the relative decrease of the 95% percentile values of PM2.5 is similar to the relative mean change in winter, however in summer,

the relative change of the high-end values of PM2.5 tends to be higher than the corresponding relative change of the mean ones.

### 3.3.4 Impact on concentrations during the episodes

In order to demonstrate, how model concentrations respond to the introduction of urban surfaces (i.e. to UCMF) during episodes of extreme air pollution and whether the modeled changes of the high-end of the distribution are in line with the model behavior

during these episodes, we plotted the "urban" (U) and "nourban" (NU) evolution of the concentrations from different model experiments during these episodes, see Fig. 9. We also included the observed values in order to see how the model accuracy changes due to UCMF.

Looking at the upper panel with ozone, it is clear that urban effects, as expected, usually increase the simulated ozone. This increase is changing day-by-day and is different in each model, but is around 5-10 $\mu gm^{-3}$ in average which is roughly

2.5-5 ppbv, i.e. very similar to the change of the 95% percentiles seen in Tab. 3. The differences between the "U" nad "NU" experiments are of course caused not only by the introduction of urbanized surfaces but also by some higher order effects (especially for secondary chemical species) that the urban canopy has on the physical properties of the air within and above this canopy, therefor it is clear that during certain conditions, $O_3$ "nourban" values can be even higher compared to the "urban" values. For $NO_2$, the effect is more unified between models and confirms the results seen in Tab. 5: $NO_2$ concentrations due

to the UCMF can be lower by up to 10 $\mu gm^{-3}$ (roughly 5 ppbv) which is, again, very close to values for the 95% percentile changes.

During the winter episode, PM2.5 is clearly decreased by UCMF in every model and the decrease lies between 5 and 10 $\mu gm^{-3}$ (largest in the WRF/CAMx experiments), which perfectly matches the interval seen for the 95% percentile change in Tab. 5. The modeled "urban" and "nourban" $NO_2$ values during the winter episodes confirm the expected UCMF too: the

"urban" values are lower by about 5-10 $\mu gm^{-3}$, roughly 2.5-5 ppbv, compared to the "nourban" case. This is, again, in line with the values for the 95% percentile change.



## 4 Discussion and conclusions

The study reveals some yet unanswered questions about the behavior of extreme air pollution concentration in reaction to the introduction of urbanized landsurfaces. It adopted multiple regional climate model and chemistry transport model combinations
and resolution to increase the robustness of the results and combined the analysis of both the long term statistical behavior of air pollution as a response to UCMF, and its instantaneous response during particular extreme air pollution events.

The general behavior of models in terms of simulating the average regional climate (within that we investigate the urban effects) is that they perform reasonable with biases within the range of other similar studies (Berg et al., 2013; Huszar et al., 2014; Karlický et al., 2018; Huszar et al., 2020). In terms of RegCM, the large overestimation of precipitation seen in Huszar
et al. (2020) is reduced by more than 50% in this study, which can be attributed to different moisture scheme used (WSM5 compared to the Nogherotto scheme). Winter temperatures have positive bias, connected probably to increased cloudiness (in connection with positive precipitation bias) and reduced thermal cooling. Giorgi et al. (2012), encountering similar biases, suggested that the heat removal from the surface towards higher levels is probably underestimated too. The seasonal temperature and precipitation biases are very similar to regional climate models studies Berg et al. (2013) and Fallmann et al. (2017), who
used similar resolution (7 km in their case) and the WSM5 microphysics too. In WRF, precipitation has a slightly higher positive bias than in RegCM experiments and this can explain the negative temperature bias in summer (via enhanced cloudiness). In general, the 6-class PLIN scheme counts with relatively high sedimentation velocities for graupel, which means stronger precipitation formation (via riming) (Hong et al., 2009) and this could contribute to the positive rain bias in WRF simulations. Gallues and Pfeifer (2008) showed that The PLIN scheme performs almost the best compared to other microphysics scheme in
WRF, it has to be however noted, that the observed biases in the model are a combined product of different parameterizations (including boundary layer, surface layer, landsurface and other processes) and so far, according to the authors knowledge, this combination was not yet adopted in WRF studies.

During the two selected high air pollution episodes, both models largely overpredict the 10-m wind speed, especially in winter. Giorgi et al. (2012) argued that the Holtslag scheme used in the RegCM setup overpredicts the vertical transport of
momentum (and scalars too) causing stronger wind over the surface. In WRF-Chem the BouLac scheme was used that was found to better represent the PBL in regimes of higher static stability compared to non-local schemes, however, it still failed to predict the wind correctly and it exhibits similar overestimation than in e.g. Tyagi et al. (2018). Similar overestimation of wind speed in WRF was reported also by Tucella et al. (2012). Another reason for wind overestimation can be related to the urban canopy models used (remember, that observational data are from urban stations) and probably the urbanization induced wind
speed decrease is even larger than resolved by the models and their urban canopy schemes. PBL heights are simulated with acceptable accuracy with some overestimation in the RegCM model, which is probably connected to the overall overestimation of vertical turbulent transport of momentum in the Holtslag scheme (Giorgi et al., 2012).

The comparison of modeled and observed pollutant concentrations reveals multiple model deficiencies. Ozone is strongly overestimated in monthly means given mainly by the nighttime positive bias (daytime values are captured reasonably). This
behavior was encountered in previous regional climate-chemistry model studies with similar setups (Zanis et al., 2011; Huszar





et al., 2016a; Karlický et al., 2017; Huszar et al., 2018a, b, 2020) and is attributed to deficiencies in nighttime chemistry and also inaccurate vertical mixing in the nocturnal boundary layer (Zanis et al., 2011). During the selected summer episode, the 8-hour ozone daily maxima are underestimated by all simulations, despite of the fact that the maximum in the average diurnal cycle is captured more accurately. This indicates that the models are unable to correctly resolve the highest ozone values.

$NO_2$ is systematically underestimated in all models and suggest that emissions are too low or at least the $NO + NO_2$ speciation of NOx emission is not correct. However, from the diurnal cycle in both summer and winter, it is clear, that the correlation with observation is high and this underlines the systematic character of the $NO_2$ average negative bias, which was similarly observed also in Huszar et al. (2016a) using very similar model configuration, or also in Tucella et al. (2012); Karlický et al. (2017) who both used WRF-Chem. The underestimation of $NO_2$ is clearly demonstrated also by the episode

figures. Our results also show that high resolution experiments are much more successful in capturing the day-to-day variation of pollutant concentrations, probably as a results of higher resolution of emissions and also a better representation of the terrain and therefor the meteorological conditions.

    PM2.5 is underestimated in our simulations, in both winter and summer (except one model set-up were overestimation occurs during winter). Huszar et al. (2016a) reported similar underestimations, which are attributable to underestimated nitrate

aerosol and black/organic aerosol, as seen also in Schaap et al. (2004) and Myhre et al. (2006). Probably, emission of the primary PM components are underestimated, similar to their precursors (e.g. $NO_2$) pointing to the large role emissions play in the overall model biases (Aleksankina et al., 2019).

    The average impact on temperature has expected magnitude in our simulations compared to previous regional scale studies conducted for European cities (Trusilova et al., 2008; Struzewska and Kaminski, 2012; Karlický et al., 2018; Huszar et al.,

2020). There is an indication that higher model resolution leads to higher impact in city centers (seen for Prague), however one must be careful with this conclusion as e.g. Huszar et al. (2020) reported large impact also at relatively low resolutions and even in our case, similar magnitude of impact can be achieved with lower resolution applied in other models (e.g. WRF-chem 9 km experiment vs. 1 km RegCM experiment). The changes in the height of the PBL are little bit higher than values in our previous regional scale study Huszar et al. (2018a) or in Wang et al. (2007); Zhu et al. (2017). They however used 3 km (and 9 km)

as their highest resolution and, evidently, resolution plays role here, as in our case, the urban canopy induced PBL increase is much larger for the high resolution inner domain compared to coarse outer domains (where the increase is less by about 50%) in both winter and summer. Summer PBL increase is higher which is an expected consequence of enhanced contribution of buoyant source of turbulence generation in urban areas (Fan et al., 2017) as a direct result of higher near surface temperatures and thus reduced stability. In case of the wind speed changes, our results confirm the expected behavior that the winds are

reduced due to the enhanced drag in urban areas. The reduction is greatest again for the high resolution experiments but the difference between low and high resolution results are rather small. Results are slightly smaller (around -1.5 $ms^{-1}$) compared to our previous study (up to 2 $ms^{-1}$ reduction) using similar experimental configuration (Huszar et al., 2020) but are large taen in the coarse resolution study of Huszar et al. (2018a). Wind decreases simulated for central European region by Struzewska and Kaminski (2012) or for China by Zhu et al. (2017) are smaller but this could again be the result of the coarser resolution

they applied.





Our results show interesting features of the urbanization induced modifications of extreme values of temperature, boundary layer height and wind-speed. During winter, smallest temperatures are more affected than the average ones, which is probably caused by larger anthropogenic heat source during winter cold days, in contrast with warm winter days, when the additional heat input is smaller causing smaller temperature increase (Varentsov et al. (2018); Karlický et al. (2018) showed that anthropogenic
heat is an important contributor to the winter urban heat island). The situation in summer is opposite and this reflects the drivers of the summer temperature increase in urban areas. Cold summer days with frequent cloudiness and limited sunshine are affected by less due to limited role of the radiation trapping. Hot summer days behave opposite: during them the role of the short wave radiative input from sun is much larger as well as the accumulation of heat due to multiple reflection and trapping in street canyons. Recently, Zhao et al. (2019) showed too that extreme temperature events (in terms of number of days with
maximum temperature $> 25°$ C) are rapidly increasing in frequency with increasing urbanization.

The boundary layer height (PBLH) changes in the low and high end of the probability distribution function show a more uniform picture: i.e. low values of PBLH change due to urbanization with a smaller magnitude than those corresponding to thick PBL, in both studied seasons. This can be explained by the dependence of the urbanization induced vertical turbulent diffusion ($K_v$) modifications on the absolute PBLH values, as shown by Huszar et al. (2020) who compared the magnitude of
$K_v$ for different turbulent parameterizations with the corresponding $K_v$ modifications due to urban landsurface. Indeed, during higher PBL characterized with stornger turbulent transport, an additional drag imposed by urban structures and heat source decreasing stability creates an increase of PBLH that is larger than the increase with weak turbulence. The dependence of the increase of boundary layer height and the absolute PBL is seen also from the diurnal cycles of $K_v$ published in this study.

In case of the wind-speed changes, a similar pattern is observed than for the PBLH. Low windspeed are modified by the
introduction urban landsurface less compared to high windspeed. Indeed, strong winds (95% percentile values) are modified by almost a factor 2 more than average wind-speeds. The reason for this is similar to PBLH changes: in case of low winds, the additional drag due to urban landsurface slows down the air motion in a lesser extent compared to high winds. This is seen also in the results of Huszar et al. (2020) showing that larger absolute windspeeds are associated with larger wind speed decrease. This is clearly visible even on the diurnal cycle of wind and its urban induced changes (Huszar et al., 2018a): largest absolute
winds coincide with the larges wind-speed decreases due to urban landsurface. A similar finding was published in Zhu et al. (2017) too.

The impact of the above discussed meteorological changes (what we call the urban canopy meteorological forcing – UCMF) on the average species concentration follows the expected patterns for both ozone, $NO_2$ and PM2.5. In case of ozone, increases are in line with a number of previous studies: e.g. Civerolo et al. (2007) modeled the maximum 8-hour ozone increases up to 6
ppbv, same as in our study. Jiang et al. (2008) and Xie et al. (2016a) found increases of $O_3$ due to rapid urbanization and the associated anthropogenic heat around 3-4 ppbv, again close to our finding. Huszar et al. (2020) calculated ozone increase due to enhanced urbanization induced turbulence up to 3-4 ppbv, however they concerned the change of seasonal average ozone (with similar changes than Jacobson et al. (2015)) which can be in general different from the change of the maximum 8-hour ozone. The resolution plays rather a minor role in the modeled magnitude of ozone changes, as seen in this study or noted by
Markakis et al. (2015) too who simulated the regional scale air quality of Paris. Around 3-4 % increase of surface ozone is





calculated by Wang et al. (2009) due to urbanization, similar to our relative mean 8-hour ozone changes (these are not directly comparable, but give at least some first estimate of the differences between these studies). Martilli et al. (2003) simulated peak ozone changes around 10-20 ppbv, which are again not comparable to our 8-hour averages, but suggest that the urban impact on extreme ozone values can reach very high numbers.

Our results showed that the peak (95% percentile) 8-hour ozone values increased due to urban meteorological effects by a little bit more than the mean values of this quantity, but this increase is not detectable in all model experiments and is seen mainly for the high resolution ones (and also for the WRF-Chem case). Jiang et al. (2008) also looked at changes of the frequency distribution of maximum 8-hour ozone and according to their results the high-end of the distribution changes by a similar magnitude than the median value, although it has to be noted that the changes due to climate change was included

too. To conclude, simulations show that urbanization contribute to extreme ozone concentrations, but this contribution is rather similar to the contribution to the mean values, or at least depend on the model set-up.

With regard to changes of $NO_2$, there is a clear decrease ranging from -2 to -6 ppbv depending on the model and resolution applied and being slightly higher for winter. The decrease is explained by increased vertical turbulent transport (Huszar et al., 2018a; Kim et al., 2015; Xie et al., 2016a) and the numbers are close to previous studies (e.g.; Sarrat et al., 2006; Struzewska

and Kaminski, 2012). Our simulations showed a very important feature, i.e. that days with extreme (95%) $NO_2$ pollution are much more affected (almost by a factor of 2) than the average days (i.e. those with average values) in both cold and warm season. This is in line with previous studies that looked at selected air pollution episodes with high NOx levels. E.g. Sarrat et al. (2006) simulated an anticyclonic situation with weak winds and significant solar radiation when high values of ozone and NOx occurred over Paris. Indeed, their results for $NO_2$ decreases are very large (more then -50 ppbv), supporting our

findings, that extreme air pollution events (for oxides of nitrogen in this case) are more influenced by the urbanization induced meteorological changes than long term average pollution.

The simulated PM2.5 response to UCMF follows the know pattern too, which means mostly decreases, that are larger in winter than in summer (about -4 and -2 $\mu gm^{-3}$ decreases for the average seasonal concentrations, respectively). The intermodel differences and those arising from different resolutions seem to play a rather minor role with stronger impacts simulated with

higher resolution RegCM experiments and with the WRF runs. The impact is stronger than in Huszar et al. (2018b) where the competition between wind induced increases and turbulence induced decreases resulted less in favor of turbulence and wind player a stronger role, similar to Huszar et al. (2020). On the other hand, a similar decrease was modeled using WRF in Kim et al. (2015) for Paris as in our study. Li et al. (2019b) found that the decrease of PM2.5 due to urbanization is mainly detectable during nighttime and attributable to increased ventilation and gas–particle phase partitioning effects favoring the gas phase. In

our simulations, there is a substantial difference between the change of the average values and those corresponding to the 95% percentile values – these later are almost 2 times higher, especially for high resolutions. This conclusion is similar to the $NO_2$ case.

It has to be noted, that in case of PM2.5 and NOx UCMF induced decreases, one can expect that the change will be higher for high absolute values, hence the peak (e.g. 95% percentile) values are more affected. To address this issue in more

detail, we looked also at the relative changes of these pollutants and they showed for summer and mainly for $NO_2$, that





the relative modifications are larger for the peak 95% values. For winter, however, the relative changes rather follow the above expectation. To conclude, urbanization contribute to $NO_2$ and PM2.5 extreme pollution negatively by decreasing their concentrations, which is shown to be stronger than the decrease encountered for the average values representing average air pollution conditions.

In summary, our paper focused on the investigation, whether extreme air pollution concentrations are affected by the urbanization induced meteorological modifications with the same magnitude as the average values, or the influence is much larger (smaller). We found that for 8-hour ozone, the influence is comparable between average and peak values – unlike for extreme $NO_2$ and PM2.5, which responded to these meteorological modifications much more pronounced compared to the change of average values. This indeed underlines the important role that urbanization and the accompanying meteorological influences play during adverse air pollution episodes.


*Code and data availability.*  The RegCM4.7 model is freely available for public use at https://gforge.ictp.it/gf/download/frsrelease/259/1845/RegCM-4.7.0.tar.gz (Giuliani, 2019). CAMx version 6.50 is available at http://www.camx.com/download/default.aspx (ENVIRON, 2018). The wrfcamx preprocessor is available from http://www.camx.com/download/support-software.aspx. WRF-Chem version 4.1 can be downloaded from https://www.acom.ucar.edu/wrf-chem/download.shtml (WRF, 2020). The RegCM2CAMx meteorological preprocessor used to convert
RegCM outputs to CAMx inputs is available upon request from the main author. The complete model configuration and all the simulated data (3-D for meteorological variables, 3-D for ozone and PM2.5 and 2-D for other chemical species) used for the analysis are stored at the Dept. of Atmospheric Physics of the Charles University data storage facilities (about 20TB) and are available upon request from the main author.

*Author contributions.*  PH provided the scientific idea, the design of the model experiments, the project coordination, and supervised the writ-
ing of the paper; PH, JK were responsible for performing the RegCM, CAMx and WRF-Chem experiments; JD, TN, KS and FS contributed to the evaluation of the results; all the authors contributed to writing the paper.

*Competing interests.*  The authors declare that they have no conflict of interest.

*Acknowledgements.*  This work has been funded by the Czech Science Foundation (GACR) project No. 19-10747Y and partly by the projects OP-PPR (Operation Program Prague – Pole of Growth)
CZ.07.1.02/0.0/0.0/16_040/0000383 "URBI PRAGENSI - Urbanization of weather forecast, air quality prediction and climate scenarios for Prague" and by projects PROGRES Q47 and SVV 2020 – Programmes of Charles University. We further acknowledge the TNO MACC-III emissions dataset, the Air Pollution Sources Register (REZZO) dataset issued by the Czech Hydrometeorological Institute and the ATEM Traffic Emissions dataset provided by ATEM (Studio of ecological models). We acknowledge the E-OBS dataset from the EU-FP6 project





UERRA (http://www.uerra.eu) and the Copernicus Climate Change Service, the data providers in the ECA&D project (https://www.ecad.eu)
and the data providers of AirBase European Air Quality data (http://www.eea.europa.eu/data-and-maps/data/aqereporting-1) and AIM (Au-
tomatic Imission Monitoring network data http://portal.chmi.cz/aktualni-situace/stav-ovzdusi/prehled-stavu-ovzdusi?l=en)





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



**Table 1.** The list of model simulations performed. The first section contains the RCM simulations that cover the whole analyzed period with the information whether urban landusurface was considered (2nd column). The second section lists the performed regional CTM experiments – here the second column provides information on the driving meteorological data (not needed in case of WRF-Chem). Finally, the third section lists the RCM/CTM simulations conducted over the two selected air pollution episodes in 2015.

| Regional Climate Model (RCM) runs | | | |
|---|---|---|---|
| Experiment | Urbanization[a] | Resolution[km] | Period |
| RegCM9U(/3U/1U)[b] | YES | 9/3/1 | 2014/12-2017/01 |
| RegCM9NU(/3NU/1NU) | NO | 9/3/1 | 2014/12-2017/01 |
| WRFchem9U | YES | 9 | 2014/12-2017/01 |
| WRFchem9NU | NO | 9 | 2014/12-2017/01 |
| Regional Chemistry Transport Model (CTM) runs | | | |
| Experiment | Driving Data | Resolution[km] | Period |
| RegCM/CAMx9U(/3U/1U) | RegCM9U(/3U/1U) | 9/3/1 | 2014/12-2017/01 |
| RegCM/CAMx9NU(/3NU/1NU) | RegCM9NU(/3NU/1NU) | 9/3/1 | 2014/12-2017/01 |
| WRFchem9U | –[c] | 9 | 2014/12-2017/01 |
| WRFchem9NU | – | 9 | 2014/12-2017/01 |
| WRF/CAMx9U | WRFchem9U | 9 | 2014/12-2017/01 |
| WRF/CAMx9NU | WRFchem9NU | 9 | 2014/12-2017/01 |
| Episodical Climate/Chemistry runs | | | |
| Experiment | Urbanization | Resolution[km] | Period (2015) |
| WRFchem9U(/3U/1U) | YES | 9/3/1 | 10/2-25/2 and 2/8-17/8 |
| WRFchem9NU(/3NU/1NU) | NO | 9 | 10/2-25/2 and 2/8-17/8 |
| WRF/CAMx9U(/3U/1U) | YES | 9/3/1 | 10/2-25/2 and 2/8-17/8 |
| WRF/CAMx9NU(/3NU/1NU) | NO | 9/3/1 | 10/2-25/2 and 2/8-17/8 |

[a]Information whether urban landsurface was considered.
[b]Simulation performed in a nested way on 9, 3 and 1 km.
[c]No driving meteorological data needed as chemistry is online coupled to the parent meteorological model





**Table 2.** Mean, 5% and 95% quantile of the urban canopy impact for models RegCM and WRF on near surface temperature (tas), the height of the boundary layer (PBLH) and 10-m wind speed (wind10m) averaged over DJF and JJA 2015-2016 for centers of 4 different cities. For Prague, values are taken from the 1 km simulations, while 9 km for the rest.

| Prague | | DJF | | | JJA | |
|---|---|---|---|---|---|---|
| | mean | 5% | 95% | mean | 5% | 95% |
| $\Delta$tas[°C] | 2.4/1.3[a] | 5.0/1.9 | 1.2/1.1 | 2.2/2.2 | 0.6/1.9 | 3.1/2.9 |
| $\Delta$PBLH[m] | 384/128 | 294/45 | 450/248 | 480/265 | 248/195 | 491/353 |
| $\Delta$wind10m[ms$^{-1}$] | -1.1/-0.5 | -0.34/-0.26 | -2.57/-1.6 | -0.64/-0.15 | -0.25/-0.29 | -1.45/-0.35 |
| **Berlin** | | | | | | |
| $\Delta$tas[°C] | 1.44/1.46 | 1.6/1.6 | 0.5/1.3 | 2.2/2.2 | 0.6/1.9 | 3.1/2.9 |
| $\Delta$PBLH[m] | 238/170 | 142/70 | 227/279 | 307/337 | 162/279 | 280/433 |
| $\Delta$wind10m[ms$^{-1}$] | -0.80/-0.74 | -0.38/-0.28 | -1.86/-2.1 | -0.46/-0.32 | -0.27/-0.30 | -0.80/-0.58 |
| **Munich** | | | | | | |
| $\Delta$tas[°C] | 1.33/1.82 | 2.89/2.65 | 0.59/2.19 | 1.12/2.2 | 0.53/1.96 | 1.76/2.83 |
| $\Delta$PBLH[m] | 132/106 | 65/29 | 144/157 | 244/270 | 122/201 | 362/367 |
| $\Delta$wind10m[ms$^{-1}$] | -0.46/-0.26 | -0.25/-0.21 | -0.77/-1.01 | -0.32/-0.11 | -0.21/-0.33 | -0.52/-0.36 |
| **Budapest** | | | | | | |
| $\Delta$tas[°C] | 1.13/1.37 | 2.60/1.02 | 0.54/1.74 | 1.2/2.4 | 0.71/2.16 | 1.40/2.97 |
| $\Delta$PBLH[m] | 132/122 | 106/67 | 268/248 | 265/336 | 132/225 | 236/509 |
| $\Delta$wind10m[ms$^{-1}$] | -0.91/-0.17 | -0.33/-0.24 | -1.92/-1.00 | -0.59/-0.20 | -0.39/-0.32 | -0.83/-0.69 |

[a]RegCM vs. WRF(-chem)

**Table 3.** Mean, 5% and 95% quantile of the urban canopy impact on JJA maximum daily 8-hour ozone (DMAX8HO3) in ppbv for different city centers. Three numbers stand for the following experiments: RegCM/CAMx, WRF/CAMx, WRFchem averaged over 2015-2016 for 4 different cities. For Prague and the RegCM/CAMx experiments, values are taken from the 1 km simulations, while 9 km for the rest.

| $\Delta$DMAX8HO3[ppbv] | mean | 5% | 95% |
|---|---|---|---|
| Prague | 2.8/2.3/2.8[a] | 2.5/3.4/2.3 | 3.8/2.9/4.3 |
| Berlin | 2.9/3.2/3.2 | 3.7/5.0/2.6 | 3.5/2.0/4.2 |
| Munich | 2.5/2.8/3.3 | 2.3/3.4/2.0 | 1.8/1.0/4.4 |
| Budapest | 2.1/2.9/3.2 | 1.7/6.2/3.7 | 2.4/1.4/4.6 |

[a]RegCM/CAMx, WRF/CAMx, WRFchem results

**Table 4.** Same as 3, but in relative change with respect to the "nourban" (NU) case in %.

| $\Delta$DMAX8HO3[%] | mean | 5% | 95% |
|---|---|---|---|
| Prague | 7/9/8[a] | 10/22/12 | 7/2/9 |
| Berlin | 10/18/10 | 22/53/14 | 7/4/9 |
| Munich | 6/10/9 | 8/21/11 | 3/2/9 |
| Budapest | 6/14/10 | 8/47/20 | 4/3/9 |

[a]RegCM/CAMx, WRF/CAMx, WRFchem results





**Table 5.** Mean, 5% and 95% quantile of the urban canopy impact on daily mean $NO_2$ and PM2.5 in ppbv and $\mu gm^{-3}$ for DJF and JJA. Three numbers stand for the following experiments: RegCM/CAMx, WRF/CAMx, WRFchem averaged over 2015-2016 for the centers of 4 different cities. For Prague and the RegCM/CAMx experiments, values are taken from the 1 km simulations, while 9 km for the rest.

| $\Delta NO_2[ppbv]$ | | DJF | | | JJA | |
|---|---|---|---|---|---|---|
| | mean | 5% | 95% | mean | 5% | 95% |
| Prague | -3.3/-3.2/-4.0 | -0.5/-0.9/-0.7 | -7.2/-4.3/-5.2 | -2.7/-3.5/-3.4 | -1.3/-1.9/-1.8 | -4.2/-4.6/-4.8 |
| Berlin | -1.9/-2.1/-4.5 | -0.5/-0.5/-0.6 | -3.4/-3.3/-5.0 | -1.7/-7.0/-7.2 | -1.0/-3.1/-2.3 | -2.5/-10.0/-12.4 |
| Munich | -1.9/-3.7/-4.5 | -0.7/-0.5/-0.5 | -1.6/-4.5/-5.9 | -1.4/-5.6/-4.7 | -0.1/-3.3/-2.3 | -1.8/-8.2/-6.4 |
| Budapest | -0.8/-4.3/-7.0 | -0.6/-0.5/-0.9 | -0.9/-5.9/-12.0 | -0.9/-5.4/-5.0 | -0.5/-2.0/-1.6 | -1.3/-9.2/-8.1 |
| $\Delta PM2.5[\mu gm^{-3}]$ | | DJF | | | JJA | |
| | mean | 5% | 95% | mean | 5% | 95% |
| Prague | -3.9/-5.0/-4.4 | -0.8/-1.2/-0.8 | -9.7/-7.5/-6.7 | -1.7/-2.3/-2.5 | -0.7/-1.2/-1.3 | -4.3/-3.4/-3.2 |
| Berlin | -0.8/-1.1/-1.4 | -0.5/-0.1/-0.1 | -1.7/-1.7/-2.7 | -0.7/-2.2/-1.4 | -0.1/-1.1/-0.6 | -3.0/-2.8/-1.5 |
| Munich | -0.2/-1.6/-1.8 | -0.1/-0.1/-0.3 | -1.3/-4.1/-3.0 | -0.4/-1.9/-1.2 | -0.1/-0.8/-0.8 | -1.8/-2.5/-1.8 |
| Budapest | -0.8/-5.5/-5.5 | -0.6/-1.4/-0.7 | -0.9/-9.8/-10.3 | -0.9/-2.5/-1.8 | -0.5/-1.5/-1.1 | -1.3/-3.9/-1.9 |

**Table 6.** Same as 5, but in relative change with respect to the "nourban" (NU) case in %.

| $\Delta NO_2[\%]$ | DJF | | JJA | |
|---|---|---|---|---|
| | mean | 95% | mean | 95% |
| Prague | -20/-18/-20 | -32/-19/-16 | -23/-38/-28 | -32/-48/-39 |
| Berlin | -12/-12/-15 | -15/-14/-12 | -13/-40/-30 | -13/-54/-47 |
| Munich | -18/-18/-15 | -8/-12/-12 | -23/-45/-30 | -23/-58/-37 |
| Budapest | -17/-23/-23 | -17/-23/-27 | -10/-42/-30 | -10/-66/-48 |
| $\Delta PM2.5[\%]$ | DJF | | JJA | |
| | mean | 95% | mean | 95% |
| Prague | -15/-12/-19 | -22/-12/-12 | -16/-23/-23 | -24/-25/-23 |
| Berlin | -4/-4/-10 | -6/-4/-10 | -8/-21/-19 | -16/-24/-19 |
| Munich | -3/-8/-11 | -6/-11/-11 | -4/-18/-19 | -12/-19/-23 |
| Budapest | -4/-15/-18 | -3/-16/-20 | -4/-22/-19 | -5/-26/-18 |

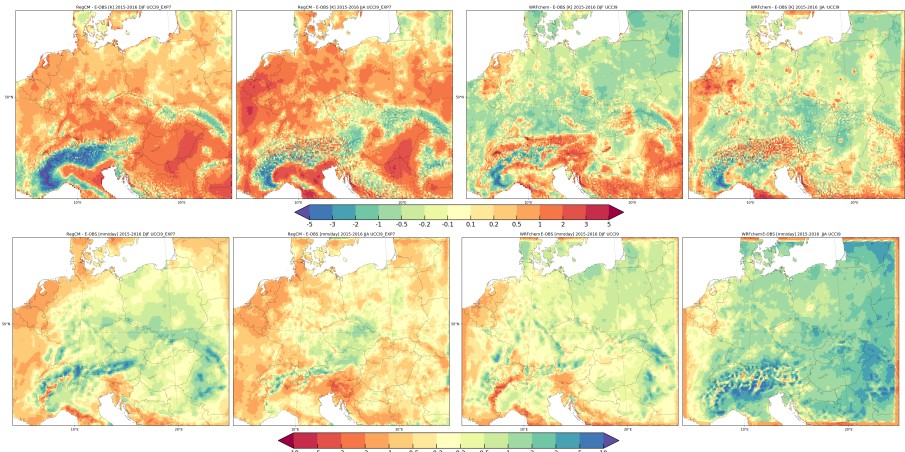

**Figure 1.** The difference between RegCM and WRF-Chem near surface temperature (upper row) and the average daily precipiation totals in mm/day (lower row) and E-OBS data for 2015-2016 DJF (1st and 3rd columns) and DJF (2nd and 4th columns) for the 9 km experiments in ° C.

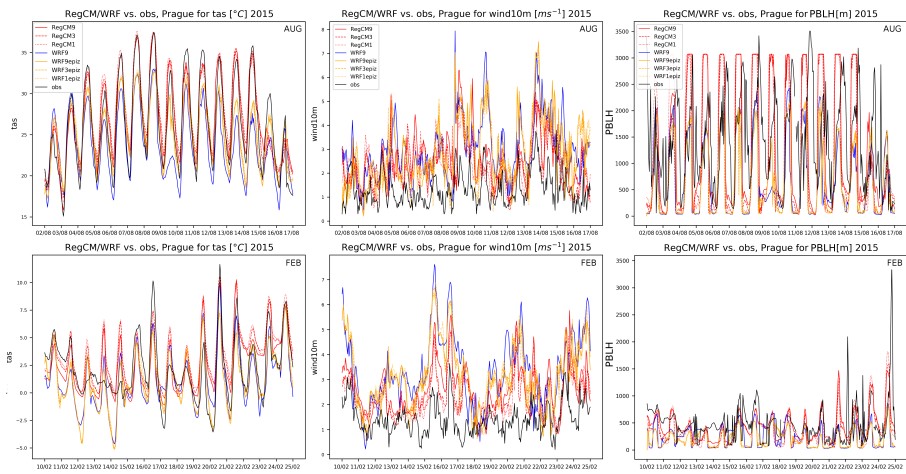

**Figure 2.** Comparison of modeled near surface temperature (tas; left), 10-m wind speed (wind10m; middle) and boundary layer height (PBLH; right) with station (average from two urban stations) data over Prague for the summer (up) and the winter period for RegCM simulations in all resolutions (9, 3 and 1 km), for the 9 km WRF-Chem simulation and for the "episodical" WRF-Chem simulations at all three resolutions (see Tab. 1).



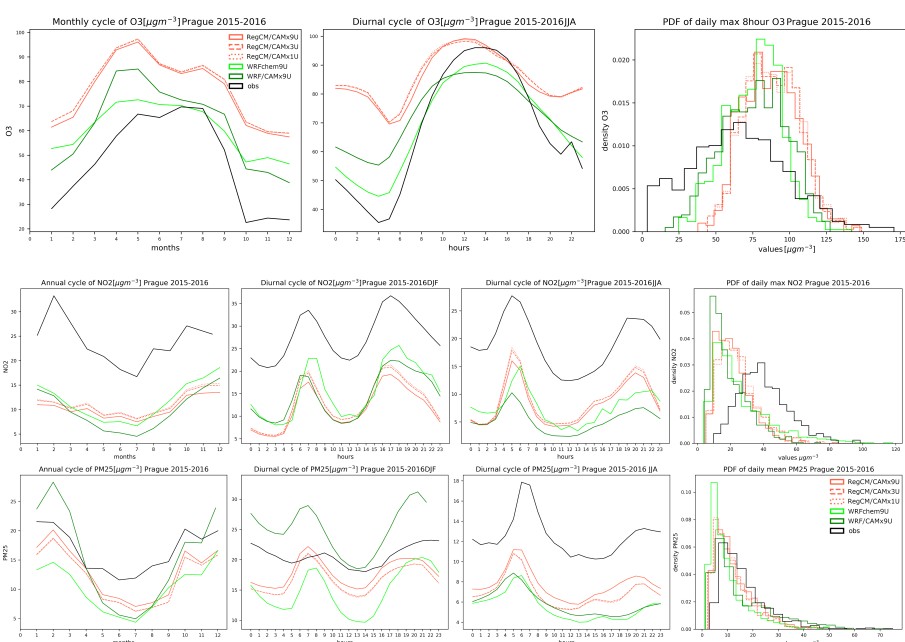

**Figure 3.** Comparison of modeled $O_3$ (up), $NO_2$ (middle) and $PM2.5$ (bottom) near surface concentrations with AirBase measurements over Prague. Three different statistics are evaluated for the 2015-2016 period: the average annual cycle, the average DJF and JJA diurnal cycle (for ozone only for JJA) and the histograms (probability density functions; PDFs) of the daily average values for the RegCM driven CAMx simulations (red), for the 9 km WRF-Chem (light green) and WRF/CAMx (dark green) "urban" (U) experiments (see Tab. 1). Observational data in black. Units in $\mu gm^{-3}$.



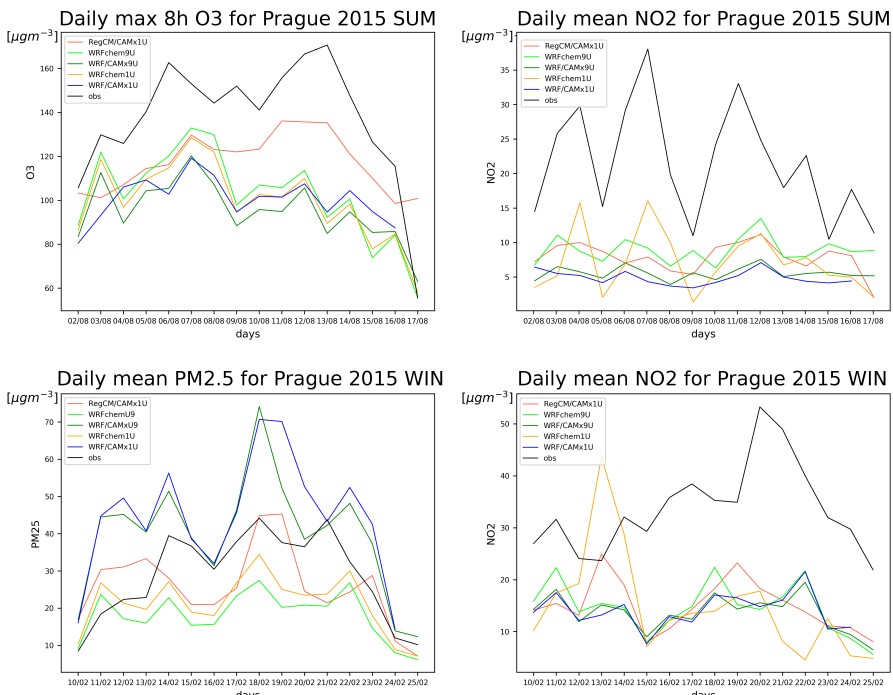

**Figure 4.** Comparison of modeled and observed maximum daily 8-hour $O_3$ and daily mean PM2.5 and $NO_2$ concentrations for the winter (up) and summer (down) period for the 1 km RegCM driven CAMx run (red), 1 km "episodical" WRF-Chem (orange) run, 1 km WRF driven "episodical" CAMx run (blue), 9 km WRF-Chem run (light green) and 9 km WRF driven CAMx run (dark green). All model results are from "urban" (U) experiements. Observational data in black. Units in $\mu gm^{-3}$.

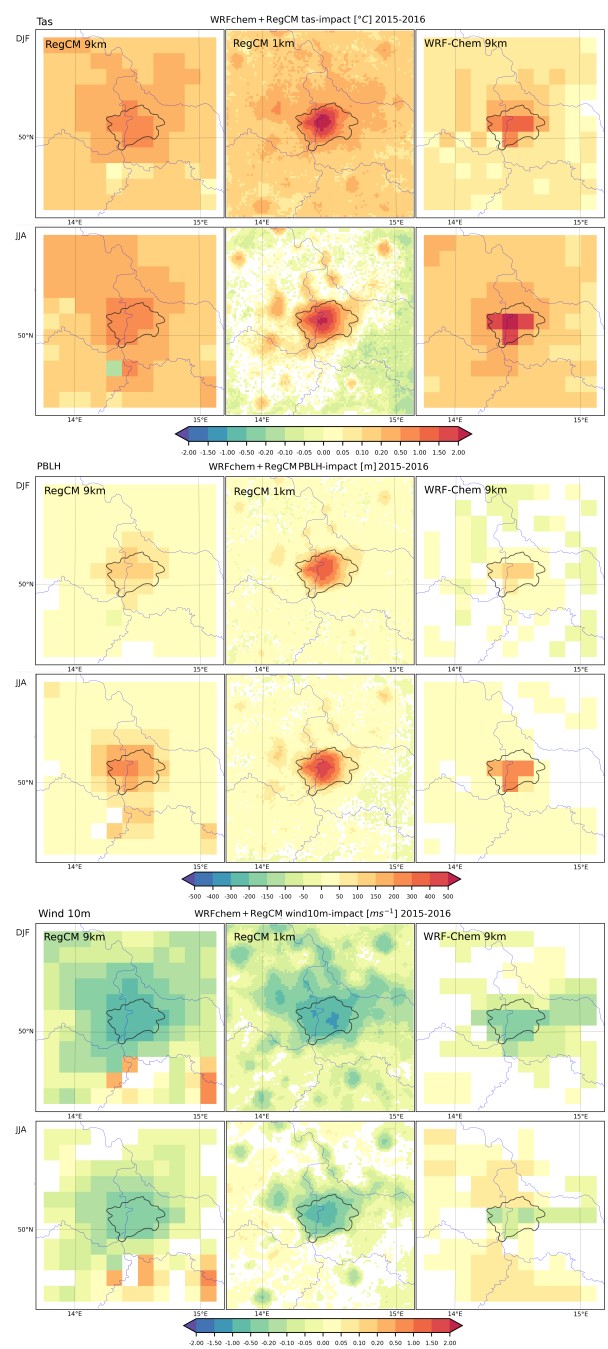

**Figure 5.** Components of the urban canopy meteorological forcing (UCMF) as the difference between "urban" (U) and "nourban" (NU) simulations for the 9 km RegCM, 1 km RegCM and 9 km WRF-Chem experiments for near surface temperature (upper panel), bounary layer height (middle panel) and 10-m wind speed (bottom panel) for the area of Prague (with plotted administrative boundaries) as 2015-2016 winter (DJF) and summer (JJA) average. Shaded areas represent statistically significant differences on the 98% level (evaluated using t-test).

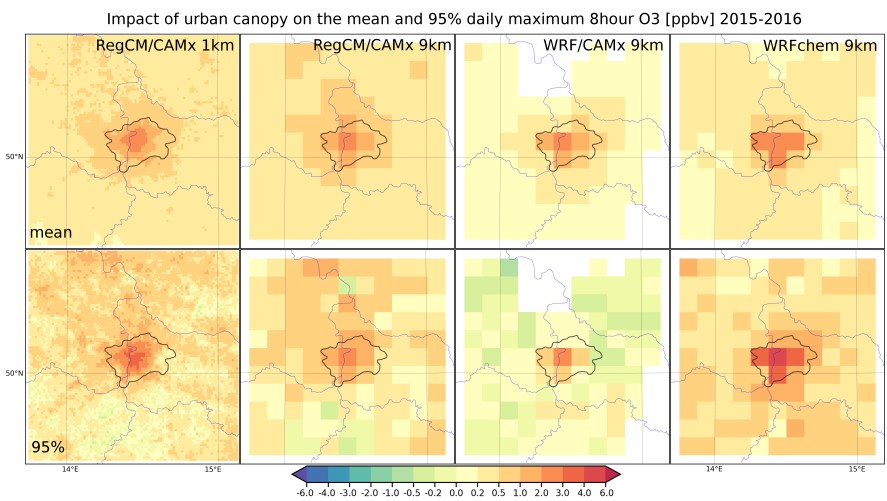

**Figure 6.** The UCMF impact on the 2015-2016 JJA mean (1st row) and the 95% percentile (2nd row) $O_3$ in ppbv for the 1 km RegCM/CAMx, 9 km regCM/CAMx, 9 km WRF/CAMx and 9 km WRF-Chem experiments as the difference between the "urban" (U) and "nourban" (NU) simulations. Shaded areas represent statistically significant differences on the 98% level (evaluated using t-test).



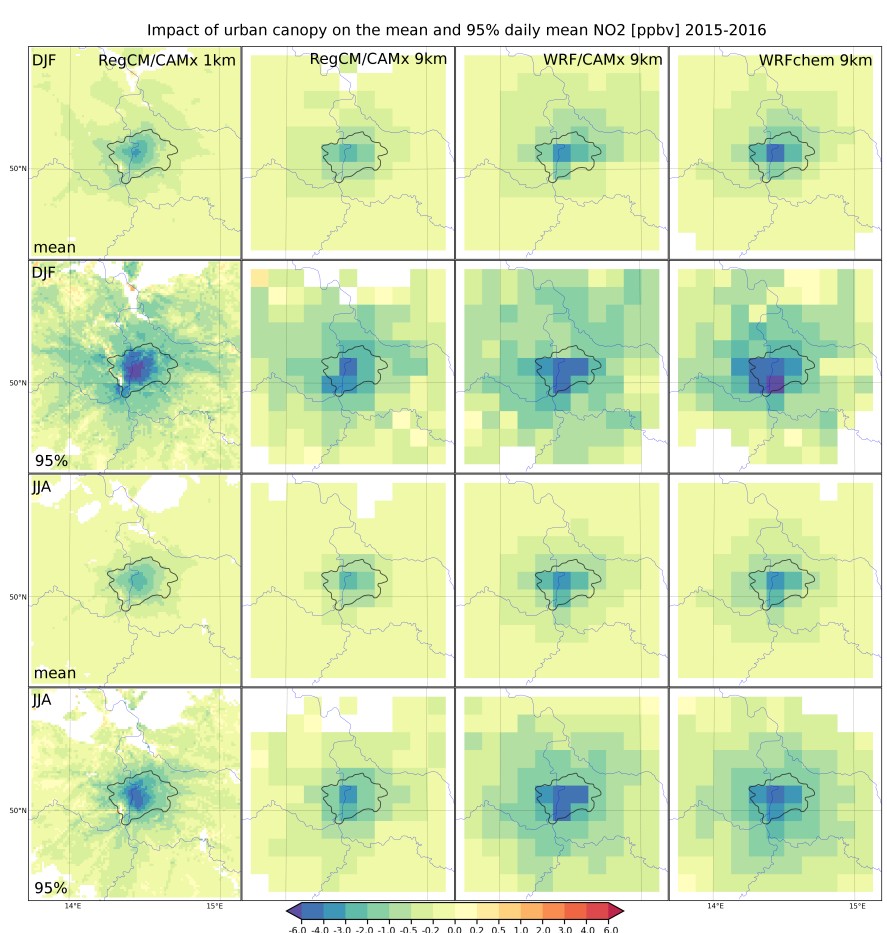

**Figure 7.** The UCMF impact on the 2015-2016 DJF and JJA mean (1st and 3rd rows) and the 95% percentile (2nd and 4th row) NO$_2$ concentrations in ppbv for the 1 km RegCM/CAMx, 9 km regCM/CAMx, 9 km WRF/CAMx and 9 km WRF-Chem experiments as the difference between the "urban" (U) and "nourban" (NU) simulations. Shaded areas represent statistically significant differences on the 98% level (evaluated using t-test).

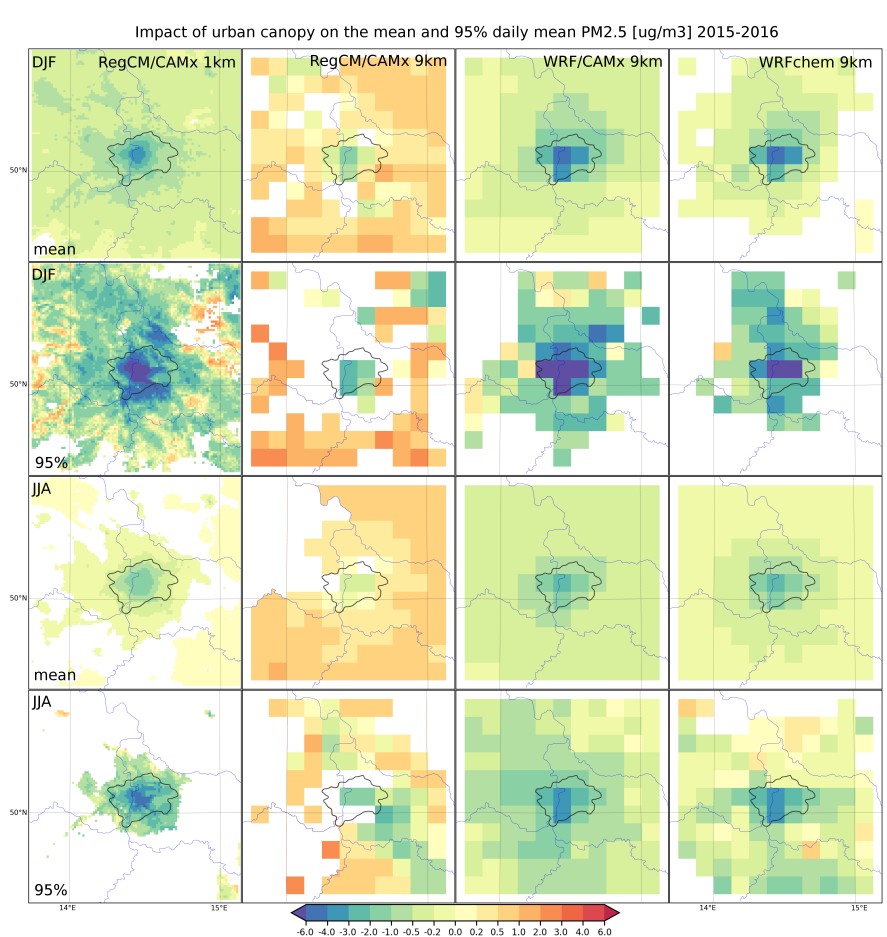

**Figure 8.** The UCMF impact on the 2015-2016 DJF and JJA mean (1st and 3rd rows) and the 95% percentile (2nd and 4th row) PM2.5 in $\mu$gm$^{-3}$ for the 1 km RegCM/CAMx, 9 km RregCM/CAMx, 9 km WRF/CAMx and 9 km WRF-Chem experiments as the different between the "urban" (U) and "nourban" (NU) simulations. Shaded areas represent statistically significant differences on the 98% level (evaluated using t-test).



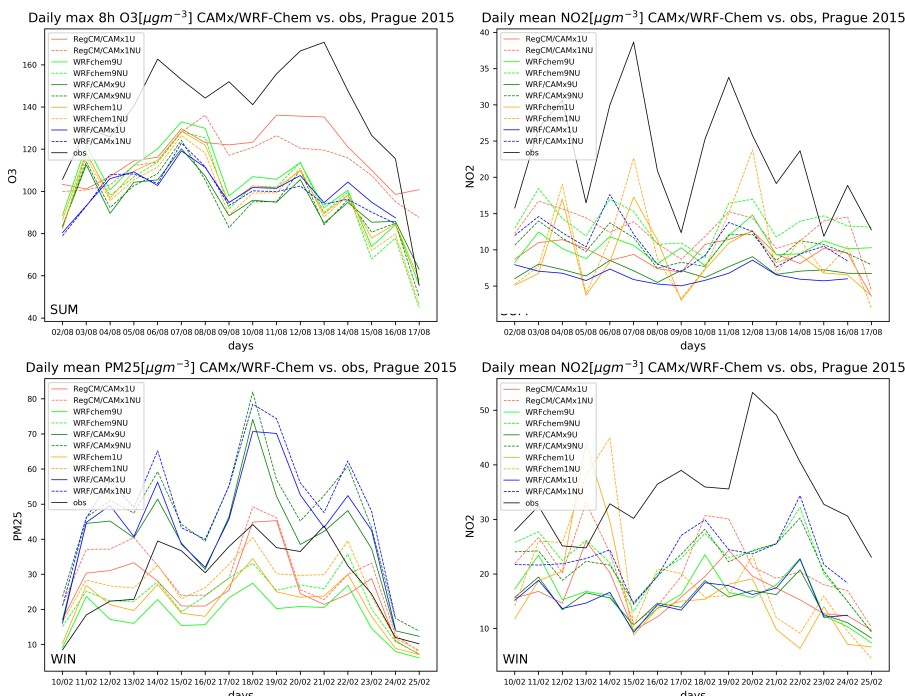

**Figure 9.** Comparison of the modeled and observed $O_3$ (left) and $NO_2$ (right) near surface concentrations for the summer high ozone epizode (top) and of the modeled and observed PM2.5 (left) and $NO_2$ (right) near surface concentrations for the winter high PM episode (bottom). Colors stand for different model simluations (1): red stands for the 1 km RegCM/CAMx, green for the 9 km WRF-Chem, dark green for the 9 km WRF/CAMx and orange for the 1 km WRF-Chem and blue for the 1 km WRF/CAMx simulations. Black stands for observations. Solid line means "urban" (U), dashed "nourban" (NU) experiment. Units in $\mu gm^{-3}$.