# Peer review of "The impact of urban land-surface on extreme air pollution over central Europe"

_Atmospheric Chemistry and Physics, 2020_

## Referee Comment (RC1) · Anonymous Referee #2 · 21 Jul 2020

General comment:

Overall, I think this is an interesting piece of work, and evidently there has been a lot of time taken to use different models, and model setups, which provides an informative comparison. I think by looking at the extreme values (5th & 95th percentiles) the authors highlight the importance of the "extremes" and the way in which they can impact air quality, rather than just looking at the mean – something which could be of use to local authorities. I think they show the impact that urbanisation can have on air quality, and which meteorological variables can further enhance this during high pollution episodes. A few things need further explanation in the methods section (described below), and I think the discussion / conclusion section needs more refining. In particular I think that there should be some mention of a comparison between overall model performance over the region. In the conclusion I think an overarching statement relating back to urban land surface / air quality would be helpful to highlight the importance of this work. There is also a problem with text size on all figures, these will have to be plotted again as they are far too small. Also, it would be helpful to have lettered/numbered plots to help make figure captions clearer. I think if these things are cleared up, I am happy to recommend this paper is accepted for publication in ACP.

Specific comments:

Page 1, Line 4: Most of the studies – makes it sound like you mean the model setups you have just described. Make this clearer that you mean studies from the literature.

1,10-14: Values helpful but percentages could help with contextualising values.

1,11 modeled -> Modelled (this is first noticed here, but occurs multiple times throughout).

2,39 know -> known

2,46-47 Sentence doesn't read well "The second influence can on the other hand". Consider rewording.

5,153 – Model setup section. There is very limited information on the domain in which you are running. I know it is central Europe, but more information should be provided. Maybe a plot showing model domain with each nest location (very common plot when running multiple nests) would be helpful here. This could also give you an opportunity to show the locations of the 4 major cities you focus on.

6, 160 – "about" / "around" seems vague. I know this varies across time, but a more quantifiable description would be helpful given that the bottom model layer is of importance in this study.

6,190 – Perhaps I couldn't see this, but there is no information given on the resolution of the anthropogenic TNO MACC-III emissions/ high res Czech emissions. What

constitutes high resolution? You are running 3 domains at high resolution but what resolution is the emission data feeding these domains? More information is needed here.

7,215 – "In THE case of WRF-Chem"

7,218 – You say that you think the overestimation is associated with a size-limited network of monitoring stations which cannot resolve local variations – how much is this effecting it? If you are using this as your explanation, why use these regridded observations at all. Can you get the individual station datasets and compare with these instead? Therefore, removing the issue of interpolation/regridding?

7,235-243 – PBL height comparisons. Are you comparing like for like here? PBLH is often calculated differently across models. Information on how the different models calculate this might be helpful

10,307/8 – Why are one set of results from the 1km domain, but all others from 9km domain.

18, 562-570 – I feel like more needs to be added to the discussion & conclusion, especially in regard to individual cities which are mentioned in Table 2-6 and in the results section. The discussion section isn't specific enough and talks about the variables (both meteorological and chemical) overall, despite large differences over the total domain. There are no concluding remarks about the different models.

31, Fig 1 – There might be some issue between the boundary conditions and the outer nest? There seems to be a "border" of higher values around the 2 most right plots (4th column). I have seen this myself when plotting WRF-Chem results and I think (can't remember 100%), but this may be to do with you plotting values outside of the actual modelled domain.

31, Fig 1 (continued) – Text size is far too small. Cannot read the title, lat/lon values or colour scale values. I would consider labelling each plot (a-h for example), because

referring to plots by upper/lower row, 1-4th column is not ideal. Also consider titling each row.

31, Fig 2 – Similar to above, text on plots far too small. Cannot see without significantly zooming in. Again, I think labelling individual plots will make the figure caption easier to understand. At the moment it is hard to follow.

32, Fig 3 – Same as above.

33, Fig 4 – Title size good here!

34, Fig 5 – Image Text size far too small again. Where you say shaded do you just mean where there is colour plotted? I.e. Where the plot is white = not statistically significant? I think the word shaded might need to be changed.

---

## Author Comment (AC1) · 13 Aug 2020

Dear Referee #2,

Thank you for your detailed review and for sharing valuable comments! We will address each of them one-by-one and our responses follow below including eventual modifications made in the revised manuscript.

General comment Overall, I think this is an interesting piece of work, and evidently there has been a lot of time taken to use different models, and model setups, which provides an informative comparison. I think by looking at the extreme values (5th & 95th percentiles) the authors highlight the importance of the "extremes" and the way in which they can impact air quality, rather than just looking at the mean – something

which could be of use to local authorities. I think they show the impact that urbanisation can have on air quality, and which meteorological variables can further enhance this during high pollution episodes. A few things need further explanation in the methods section (described below), and I think the discussion / conclusion section needs more refining. In particular I think that there should be some mention of a comparison between overall model performance over the region. In conclusion I think an overarching statement relating back to urban land surface / air quality would be helpful to highlight the importance of this work. There is also a problem with text size on all figures, these will have to be plotted again as they are far too small. Also, it would be helpful to have lettered/numbered plots to help make figure captions clearer. I think if these things are cleared up, I am happy to recommend this paper is accepted for publication in ACP.

Authors response: We greatly appreciate that the reviewer considers our paper as interesting and worth publishing. Following his comments, in the revised manuscript we will provide more information on the model setup, model performance over the region, and also provide some final overarching statements and improve the presentation quality regarding figures.

Specific comments

Page 1, Line 4: Most of the studies – makes it sound like you mean the model setups you have just described. Make this clearer that you mean studies from the literature.

Authors' response: we rephrased the statement to make clear that this refers to the literature which dealt with the urban canopy meteorological forcing on air pollution.

1,10-14: Values helpful but percentages could help with contextualising values.

Authors' response: we added the percentage values too.

1,11 modeled -> Modelled (this is first noticed here, but occurs multiple times throughout).

Authors' response: All occurrences corrected to "modelled" (i.e. British spelling).

2,39 know -> known

Authors' response: corrected.

2,46-47 Sentence doesn't read well "The second influence can on the other hand". Consider rewording.

Authors' response: we reworded the sentence in the revised manuscript

5,153 – Model setup section. There is very limited information on the domain in which you are running. I know it is central Europe, but more information should be provided. Maybe a plot showing model domain with each nest location (very common plot when running multiple nests) would be helpful here. This could also give you an opportunity to show the locations of the 4 major cities you focus on.

Authors' response: In the revised manuscript, we added in the parentheses for each resolution an approximate geographic extent and we added also a new figure (Fig. 1) which shows the nesting structure of the domains with the resolved terrain including the cities analyzed in the study.

6, 160 – "about" / "around" seems vague. I know this varies across time, but a more quantifiable description would be helpful given that the bottom model layer is of importance in this study.

Authors' response: We changed this to approximately 30 m. Indeed, the model layers are defined as sigma layers and their thickness varies with temperature according to the barometric formula. However, the lowermost layer changes only between 28 to 32 m, so we consider to use the word "approximately" as appropriate.

6,190 – Perhaps I couldn't see this, but there is no information given on the resolution of the anthropogenic TNO MACC-III emissions/high res Czech emissions. What constitutes high resolution? You are running 3 domains at high resolution but what resolution is the emission data feeding these domains? More information is needed here.

Authors' response: The TNO data have a resolution of about 6km x 6km which is sufficient for describing emissions at 9 km and partially at 3km. For the area of Czech republic, the REZZO and ATEM data were used which provide emission on irregular shapefiles, as lines (for road transportation), points (point sources) and different irregular shapes (counties) that have characteristic geometric size from a few 100m to 1-2 km. Emissions defined over these irregular shapes are then spatially interpolated to the model's Cartesian grid. In the revised manuscript, we provided this information in more detail.

7,215 – "In THE case of WRF-Chem"

Authors' response: corrected.

7,218 – You say that you think the overestimation is associated with a size-limited network of monitoring stations which cannot resolve local variations – how much is this effecting it? If you are using this as your explanation, why use these regridded observations at all. Can you get the individual station datasets and compare with these instead? Therefore, removing the issue of interpolation/regridding?

Authors' response: our intention here was to evaluate the overall spatial model performance without focus on particular cities, i.e. whether there are systematic biases in the regional modelled fields. The E-OBS is a well-known dataset and the latest version was used here ensuring high quality of data. However, we admit that urban areas are not so well represented in E-OBS which cannot resolve the higher temperatures over cities. This led to overestimation seen especially for the WRF model.

7,235-243 – PBL height comparisons. Are you comparing like for like here? PBLH is often calculated differently across models. Information on how the different models calculate this might be helpful

Authors' response: With this figure, we intend to provide only a rough comparison of the typical PBLH values from measurements vs. model values. The BouLac scheme

in WRF diagnoses PBLH as the height where the prognostic TKE reaches a small value. In the Holtslag scheme, PBLH is determined where the bulk Richardson number reaches a critical value. There are studies (e.g. Wang et al., 2014; Guttler et al., 2014) that argue that the non-local diagnostic Holtslag scheme results in stronger mixing and higher mixing than instead using a prognostic TKE (turbulent kinetic energy) based approach (like the BouLac). This explains in general the higher values of PBLH in RegCM (Holtslag) compared to WRF (BouLac) with the measured values often lying in between. Consequently, we can say that the measured values lie well within the intermodel spread. In the revised manuscript, we added a comment on this issue with the mentioned references (also in the Discussion of the results.)

10,307/8 – Why are one set of results from the 1km domain, but all others from 9km domain.

Authors' response: we admit that this was not clearly formulated in the text. The domains are nested telescopically for Prague, so this city is of main focus and the results for it are from the 1 km runs. All other cities are outside of the 1 km domain and except Munich, also outside of the 3 km domain. Therefore, for these cities, the 9 km runs provide results (for Munich, we have chosen the 9 km domain as it is right at the edge of the 3 km domain and we want to avoid some boundary effects).

18, 562-570 – I feel like more needs to be added to the discussion & conclusion, especially in regard to individual cities which are mentioned in Table 2-6 and in the results section. The discussion section isn't specific enough and talks about the variables (both meteorological and chemical) overall, despite large differences over the total domain. There are no concluding remarks about the different models.

Authors' response: we extended our discussion by commenting on the relatively large differences in both meteorology and chemistry between individual cities and models. We formulated several reasons and the impact of resolution is detailed too. We extended the concluding remarks and included some overarching remarks that put our

research in a wider scope with recommendations for aiming future research.

31, Fig 1 – There might be some issue between the boundary conditions and the outer nest? There seems to be a "border" of higher values around the 2 most right plots (4th column). I have seen this myself when plotting WRF-Chem results and I think (can't remember 100%), but this may be to do with you plotting values outside of the actual modelled domain.

Authors' response: this is related to the buffer cells which serve to relax the boundary condition values towards the model domain interior. This means that close to boundaries, model values tend to mimic boundary values which results in a "frame"-like pattern seen in this figure. We added a note regarding this to the figure caption as well as to the corresponding paragraph.

31, Fig 1 (continued) – Text size is far too small. Cannot read the title, lat/lon values or colour scale values. I would consider labelling each plot (a-h for example), because referring to plots by upper/lower row, 1-4th column is not ideal. Also consider titling each row.

Authors' response: we enlarged the titles of individual subfigures to make clear which stand for which model/variable/season and also increased other labels size (axes, colobar). Labelling subfigures with a-h is a standard element of the typesetting process (including the necessary modifications in the figure caption) and will be done by the publisher once the manuscript is accepted for publication.

31, Fig 2 – Similar to above, text on plots far too small. Cannot see without significantly zooming in. Again, I think labelling individual plots will make the figure caption easier to understand. At the moment it is hard to follow.

Authors' response: same as above.

32, Fig 3 – Same as above.

Authors' response: same as above.

33, Fig 4 – Title size good here!

Authors' response: same as above.

34, Fig 5 – Image Text size far too small again. Where you say shaded do you just mean where there is colour plotted? I.e. Where the plot is white = not statistically significant? I think the word shaded might need to be changed.

Authors' response" we increased the size of the titles, colorbars and other script. However, more increase is not necessary as figure (as all other) has a high dpi (dots per inch) resolution. Shades here means statistical significant values. We changed the formulation to "white color means statistically insignificant results".

References:

Güttler, I., Brankovic, Č., O'Brien, T.A., Coppola, E., Grisogono, B. and Giorgi, F.: Sensitivity of the regional climate model RegCM4.2 to planetary boundary layer parameterization, Clim. Dyn., 43, 1753-1772, doi:10.1007/s00382-013-2003-6, 2014.

Wang, Z. Q., Duan, A. M. and Wu, G. X.: Impacts of boundary layer parameterization schemes and air-sea coupling on WRF simulation of the East Asian summer monsoon, Science China: Earth Sciences, 57, 1480-1493, doi:10.1007/s11430-013-4801-4, 2014.

---

## Referee Comment (RC2) · Anonymous Referee #3 · 15 Aug 2020

This paper presents an analysis of the impact of representing the urban canopy in chemistry-climate or air quality simulations. A regional climate and a weather model are used, along with 2 chemistry configurations of those models. In addition, multiple horizontal resolutions are simulated. The results illustrate the value of including a representation of urban canopy in air quality and climate models so as to more accurately simulate air pollution extremes. I find the paper generally acceptable for publication, but have a few recommendations for improvements.

I feel a bit more discussion of the role of chemistry in controlling the NOx and O3 (and PM2.5) concentrations would be appropriate. Two different chemistry and aerosol schemes are used; a discussion of how they differ, and showing results that clearly illustrate if they give similar or different results would be interesting. Also, ozone forma-

tion is affected by temperature, solar radiation (cloud cover); how do the meteorology changes caused by the urban canopy affect the chemistry?

A section on the observations used to evaluate the model is needed. What is the accuracy of the observations? Where are they located?

A description of how PM2.5 was determined from the model results is needed.

The line plots showing all the numerous model results are very difficult to read. It would be helpful to have separate plots to illustrate specific differences, such as 1 set of plots to show the difference in resolution for one model, and another set of plots to show multiple models at 1 resolution. Or find some other way to illustrate those model differences (e.g., biases, bar charts of mean bias, correlations, etc.).

I found the Discussion section a bit difficult to read. It would be helpful if the figures more clearly illustrated the points discussed in this section and were referred to at appropriate points. It would also be helpful to have subsections in the Discussion, perhaps separating the findings related to the urban vs no-urban simulations, differences due to model resolution, differences due to chemistry, for example.

Minor/Technical comments

l. 16 (and elsewhere): 5% percentiles is usually written 5th percentile.

l. 124-127: define TUV and MEGAN acronyms

l. 187: "Chemical boundary conditions for the outer domains were taken from the CAM-chem data (Lamarque et al., 2012)." Be more specific about where the boundary conditions come from. I do not know of any archived results from the Lamarque et al. 2012 paper. If they are from the results provided by NCAR they should be referenced as described on: https://wiki.ucar.edu/display/camchem/CESM2.1%3ACAM-chem+as+Boundary+Conditions If you ran your own simulations, the details of that should be given.

l.204: perhaps more details of how MEGAN was run could be included - which vegetation map, which meteorology data, or is MEGAN online in the model?

l. 487: "large taen" -> larger than?

l. 562-4: I don't follow this statement. Should "decreasing" be "increasing"?

Additional proof-reading is needed. There are a number of grammar errors and typos.

---

## Author Comment (AC2) · 24 Aug 2020

Authors response on: "The impact of urban land-surface on extreme air pollution over central Europe" (acp-2020-399)

By Peter Huszar et al., 2020

Dear Referee #3,

Thank you for your detailed review and for sharing your comments with us! We will address all of them and our one-by-one responses follow below including the modifications made in the revised manuscript.

General comment

I feel a bit more discussion of the role of chemistry in controlling the NOx and O3 (and PM2.5) concentrations would be appropriate. Two different chemistry and aerosol schemes are used; a discussion of how they differ, and showing results that clearly illustrate if they give similar or different results would be interesting. Also, ozone formation is affected by temperature, solar radiation (cloud cover); how do the meteorology changes caused by the urban canopy affect the chemistry?

Authors response: Our paper is based on previous research made to investigate the role the urban canopy plays in controlling the city scale meteorological conditions and consequently the chemistry and transport of pollutants. The mutual links between meteorology and gas-phase chemistry (NOx-ozone) in urban areas is detailed in Huszar et al.(2018a), whereas in Huszar et al.(2018b) we extended this analysis to primary and secondary aerosols. Finally in Huszar et al.(2020), after identifying that vertical eddy transport plays the most important role, we focused on this aspect of the urban canopy meteorological forcing on chemistry. All three studies give robust results on which fractional processes play a role in the modelled differences in concentrations after introducing urban landsurfaces, e.g. in case of NOx, the most important effect is the eddy removal from lower model layer while for ozone, increased urban temperatures play role too by enhancing dry deposition and the NO+O3 reaction. Furthermore, these results are in line with previous similar studies for both European and other urban areas (these studies are referred to in the Introduction and also in the Discussion). Nevertheless, to make the paper more self-explanatory, we included in the Discussion section a few notes on which processes in urban areas control the NOx-O3 and aerosol chemistry and transport (based on the findings in our previous papers.) Regarding the inter-model differences in chemistry and aerosol modules, we admit that our discussion lacks to give more detail on this issue, however our study did not intend to present itself as a model comparison study and here more models are used to solely increase the robustness of the results. This is also true for showing results from different resolutions. Our results from individual models and grid resolutions are qualitatively same and very close to each other quantitatively proving the urban canopy meteorological

effects on chemistry manifest themselves in a similar way in different models and they are not an artificial feature of a selected model or resolution. We could certainly add many comments on the reasons for some of the modeled differences between models but this would drift the focus of the paper too much from what it intends to present.

A section on the observations used to evaluate the model is needed. What is the accuracy of the observations? Where are they located?

Authors' response: We provided a new section within the Model validation subsection which describes all the measured data in detail that are used in the model validation, including their resolution and the data source.

A description of how PM2.5 was determined from the model results is needed.

Authors' response: We included a sentence how PM2.5 is obtained from model output (in WRF-Chem these are directly available, whereas in CAMx they have to be calculated as a sum of all primary and secondary aerosol).

The line plots showing all the numerous model results are very difficult to read. It would be helpful to have separate plots to illustrate specific differences, such as 1 set of plots to show the difference in resolution for one model, and another set of plots to show multiple models at 1 resolution. Or find some other way to illustrate those model differences (e.g., biases, bar charts of mean bias, correlations, etc.).

Authors' response: As our study is primarily intended to show an intermodel comparison at different resolutions, we limited on Figure 2. (Fig. 3 in the revised manuscript) the presentation to the innermost 1 km domain only, except for the 2 yr WRF-Chem run made only at 9 km (blue line). Now the figure's message is more clear. Presenting results for the 1 km only is justified also by the fact that the differences between different resolutions with the same model are very small. We also increased the font in every figure to make them easier to read (a suggestion from the other reviewer).

I found the Discussion section a bit difficult to read. It would be helpful if the figures

more clearly illustrated the points discussed in this section and were referred to at appropriate points. It would also be helpful to have subsections in the Discussion, perhaps separating the findings related to the urban vs no-urban simulations, differences due to model resolution, differences due to chemistry, for example.

Authors' response: We divided the Discussion and conclusion section into three parts, the first one discussing the model validation results, the second and third then discussing the impact on meteorological conditions and average/extreme air pollution. We also added references to figures and tables, results from which are discussed.

Minor/Technical comments

l. 16 (and elsewhere): 5% percentiles is usually written 5th percentile.

Authors' response: corrected through the entire manuscript.

l. 124-127: define TUV and MEGAN acronyms

Authors' response: defined.

l. 187: "Chemical boundary conditions for the outer domains were taken from the CAM-chem data (Lamarque et al., 2012)." Be more specific about where the boundary conditions come from. I do not know of any archived results from the Lamarque et al. 2012 paper. If they are from the results provided by NCAR they should be referenced as described on: https://wiki.ucar.edu/display/camchem/CESM2.1%3ACAMchem+as+Boundary+Conditions If you ran your own simulations, the details of that should be given.

Authors' response: We used the results provided by NCAR, so we changed the references to those offered by the UCAR wiki page (Emmons et al.2020 and Buchholtz et al.2019).

l.204: perhaps more details of how MEGAN was run could be included - which vegetation map, which meteorology data, or is MEGAN online in the model?

Authors' response: We included a description of the megan input data (plant functional types, leaf area index data, emission factor maps). The meteorology used to run MEGAN is taken from the RegCM and WRF runs except the WRF-Chem experiments where biogenic emissions are computed online. This is now clarified in the text.

l. 487: "large taen" -> larger than? Authors' response: corrected.

l. 562-4: I don't follow this statement. Should "decreasing" be "increasing"?

Authors' response: Decrease is correct as we refer to the separate impact of urban landsurface only (as in the entire manuscript), Of course, taking emission into account would completely change the narrative due to the large positive impact of urban emissions. But this is not the focus of this paper and this is clearly stated in the Introduction.

Additional proof-reading is needed. There are a number of grammar errors and typos.

Authors' response: We corrected some typos found in the manuscript. Further proof-reading will be conducted by the Copernicus publishing office if the paper is accepted for publication.

References:

Huszar, P., Karlicky, J., Belda, M., Halenka, T. and Pisoft, P.: The impact of urban canopy meteorological forcing on summer photochemistry, Atmos. Environ., 176, 209-228, https://doi.org/10.1016/j.atmosenv.2017.12.037, 2018a.

Huszar, P., Belda, M., Karlicky, J., Bardachova, T., Halenka, T., and Pisoft, P.: Impact of urban canopy meteorological forcing on aerosol concentrations, Atmos. Chem. Phys., 18, 14059-14078, https://doi.org/10.5194/acp-18-14059-2018, 2018b.

Huszar, P., Karlicky, J., Doubalová, J., Sindelářová, K., Nováková, T., Belda, M., Halenka, T., Zák, M, and Pišoft, P.: Urban canopy meteorological forcing and its impact on ozone and PM2.5: role of vertical turbulent transport, Atmos. Chem. Phys., 20, 1977-2016, https://doi.org/10.5194/acp-20-11977-2020, 2020.